# SiGra: single-cell spatial elucidation through an image-augmented graph transformer

Ziyang Tang [1], Zuotian Li [2,3], Tieying Hou[4], Tonglin Zhang[5], Baijian Yang [1] ✉, Jing Su [2] ✉ & Qianqian Song [6,7] ✉

Recent advances in high-throughput molecular imaging have pushed spatial transcriptomics technologies to subcellular resolution, which surpasses the limitations of both single-cell RNA-seq and array-based spatial profiling. The multichannel immunohistochemistry images in such data provide rich information on the cell types, functions, and morphologies of cellular compartments. In this work, we developed a method, single-cell spatial elucidation through image-augmented Graph transformer (SiGra), to leverage such imaging information for revealing spatial domains and enhancing substantially sparse and noisy transcriptomics data. SiGra applies hybrid graph transformers over a single-cell spatial graph. SiGra outperforms state-of-the-art methods on both single-cell and spot-level spatial transcriptomics data from complex tissues. The inclusion of immunohistochemistry images improves the model performance by 37% (95% CI: 27−50%). SiGra improves the characterization of intratumor heterogeneity and intercellular communication and recovers the known microscopic anatomy. Overall, SiGra effectively integrates different spatial modality data to gain deep insights into spatial cellular ecosystems.

Recent advances in spatial molecular imaging have allowed for the examination of the spatial landscapes and transcriptional profiles of complex tissues at subcellular resolution[1–3]. The interrogation of the spatial locations and gene expression of individual cells within a tissue aid in understanding the spatial heterogeneity of cell-to-cell communication and cell interactions with the surrounding environment, which are crucial for understanding disease pathology. Current commercially available technologies for single-cell spatial profiling, such as the NanoString CosMx™ Spatial Molecular Imager (SMI)[4] and the Vizgen MERSCOPE/MERFISH platforms[5,6], are capable of accurately capturing the locations of targeted transcripts, cell locations, and cell boundaries, accompanied by multichannel immunohistochemistry (IHC) images. For example, NanoString CosMx™ is capable of simultaneously assaying up to 1000 genes[4] and 100 k to 600 k cells per slide,

dramatically exceeding current single-cell omics technologies. Therefore, emerging single-cell spatial transcriptomics (SCST) commercial platforms are revolutionizing current spatial biology research, promising to spatially and functionally reveal complex architectures within tissues and furthering our insights into the mechanisms underlying disease at unprecedented resolution[7–9].

The emerging SCST multimodal data provide new opportunities for accurately identifying spatial domains, which is crucial for revealing and functionally annotating the cellular anatomy of complex tissues. Existing methods for deciphering spatial cell clusters, such as Seurat[10] and the Louvain clustering-based Scanpy[11] method, still rely on clustering methods for nonspatial single-cell RNA-seq data and only take gene expression data as input. Other methods have been developed to include spatial information to improve the identification of

[1]Department of Computer and Information Technology, Purdue University, Indiana, USA. [2]Department of Biostatistics and Health Data Science, Indiana University School of Medicine, Indiana, USA. [3]Department of Computer Graphics Technology, Purdue University, Indiana, USA. [4]Department of Pathology and Laboratory Medicine, Indiana University School of Medicine, Indiana, USA. [5]Department of Statistics, Purdue University, Indiana, USA. [6]Department of Cancer Biology, Wake Forest University School of Medicine, North Carolina, USA. [7]Department of Health Outcomes and Biomedical Informatics, College of Medicine, University of Florida, Florida, USA. ✉e-mail: byang@purdue.edu; su1@iu.edu; qsong@wakehealth.edu

spatial regions. For example, stLearn[12] leverages the gene expression of neighbouring spots and tissue image features to identify spatially distributed clusters. BayesSpace[13] enables spatial clustering through a Bayesian statistical method with joint analyses of the gene expression matrix and spatial neighbourhood information. SpaGCN[14] identifies spatial regions using a graph convolutional network, with the spatial graph constructed from gene expression and histology information. Although these methods show their capability in spatial clustering, the power of different modalities within single-molecule spatial imaging profiles has not been fully unleashed to achieve desirable performance.

In addition to domain recognition, the enhancement of spatial gene expression data also presents a significant challenge. Although great progress has been made in spatial technologies, major problems such as missing values, data sparsity, low coverage, and noise[2,15] encountered in spatial transcriptomics profiles impede the effective testing and elucidation of biological insights. Meanwhile, multichannel spatial images in single-cell spatial data consist of high-resolution, high-content features detected in the tissue, such as cell types, functions, and morphologies of cellular compartments, as well as the spatial distributions of cells. Incorporating such imaging features into transcriptomics data processing will help address the challenges of missing values and data noise. Moreover, as the spatial relations between an individual cell and its neighbouring cells can be naturally represented with a spatial adjacency graph, graph-based artificial intelligence is promising for spatial data modelling. Notably, graph-based models enhanced with attention mechanisms[16], such as the graph attention network (GAT) and graph convolutional transformer models[17,18], have demonstrated remarkable advancements and yielded significantly improved outcomes.

In this study, we developed the SiGra method, i.e., single-cell spatial elucidation through an image-augmented graph transformer, to decipher spatial domains and enhance spatial signals simultaneously. SiGra can utilize multimodalities, including multichannel images of cells and their niches, address technological limitations and achieve augmented spatial profiles. SiGra accurately recovers missing information in spatial gene expression, uncovers cellular dynamics, and reveals the spatial architecture of cellular heterogeneity within tissues. Through extensive and quantitative benchmarking with existing methods on multiple datasets, including both single-cell level and spot-level spatial data generated by different platforms, SiGra demonstrates superior performance in terms of spatial domain identification, latent embedding, and data denoising. Overall, SiGra will contribute to uncovering the complex spatial architecture within heterogeneous tissues and facilitate the acquisition of biological insights. SiGra is open-source software and is available at https://github.com/QSong-github/SiGra, with detailed tutorials demonstrating its applications to different spatial transcriptomics platforms. The web interface of the SiGra Viewer (http://sigra.sulab.io) enables users to explore the enhanced data in uniform manifold approximation and projection (UMAP) figures and spatial domains.

## Results

### Overview of the SiGra method
The SiGra method includes (1) the graph representation of the original spatial transcriptomics data (Fig. 1a) and (2) the hybrid graph transformer model to elucidate the spatial patterns and enhance the raw gene expression data (Fig. 1b).

The state-of-the-art SCST data consist of: (1) multichannel images of biomarkers for cell types (e.g., pancytokeratin or PanCK staining for tumour cells, CD3 for T cells, and CD45 for leucocytes) and cell compartments (e.g., DAPI staining for cell nuclei and CD298 staining for cell membranes). For each staining channel, a high-content greyscale image is assembled from a series of field-of-view (FOV) images; (2) the vendor-provided cell segmentation results such as the coordinates of

cell centroids and the hull of cell boundaries; and (3) the cell-level summarization of gene expression according to the coordinates of each detected transcript and the cell boundary identified from cell segmentation.

In SiGra, the single-cell spatial graph is constructed based on the spatial centroids of detected cells, with each node representing a cell and each edge representing two neighbouring cells (Euclidian distance shorter than 14–16 μm). Each node/cell within the spatial graph is accompanied by multimodal data (images and gene expression) extracted from the original spatial profiles. Specifically, for each cell, an image of 21.6 μm by 21.6 μm centred at the cell centroid is cropped from each immunohistochemistry (IHC) image. For example, as NanoString CosMx data consist of five channels (DAPI, PanCK, CD45, CD3, and CD298), each cell is associated with five single-cell images. In this way, SiGra achieves the graphical representation of spatial profiles, i.e., the single-cell spatial graph with each located cell's multichannel images and gene expression.

The SiGra model comprises three graph transformer-based encoder-decoders (imaging, transcriptomics, and hybrid) with an attention mechanism (Fig. 1b) to incorporate the single-cell multimodal data for simultaneous data enhancement and spatial domain recognition. Regarding the imaging encoder-decoder, with a cell $i$ represented by node $v_i$, an array of single-cell IHC images $M_i$ is converted to a vector $x_i$ and projected to the latent space as $z_{M,i}$ through multihead graph transformer layers (Supplementary Fig. 1a, Methods). This latent imaging feature $z_{M,i}$ then is used to reconstruct the gene expression profile $\hat{g}_{M,i}$ of cell $i$. For the transcriptomics encoder-decoder, the same architecture is used for the latent representation ($z_{g,i}$) and the reconstruction ($\hat{g}_{g,i}$) of the original expression $g_i$ in cell $i$. For the hybrid encoder-decoder, the latent imaging features $z_{M,i}$ and the latent expression features $z_{g,i}$ are concatenated and projected as a hybrid feature $z_{h,i}$, which is used to reconstruct the gene expression $\hat{g}_{h,i}$ of cell $i$. Imaging and gene expression features of neighbouring cells, represented as neighbour nodes $v_j \in \mathcal{N}(v_i)$ in the spatial graph, are also used as the input for graph transformers so that the spatial cellular information is aggregated into the model.

SiGra learns the reconstructed gene expression via a self-supervised loss that combines the mean square error (MSE) from gene embedding $L_{M,i}$, image embedding $L_{g,i}$ and combined embedding $L_{h,i}$, with the loss function $L = \sum_{i=1}^{N} \lambda_1 L_{M,i} + \lambda_2 L_{g,i} + L_{h,i}$, where the hyperparameters $\lambda_1, \lambda_2 \geq 0$, and $N$ is the total cell number. After training, SiGra outputs the hybrid reconstruction $\hat{g} = \{\hat{g}_{h,i}\}$ as the final enhanced expression profile. The latent representation, $z = \{z_{h,i}\}$, of the original SCST data is used for spatial data clustering.

With the introduced multihead attention mechanism in graph transformer layers, SiGra adaptively updates the contributions of neighbouring cells $\{v_j\}$ to cell $v_i$ by aggregating and propagating the extracted image features and the gene expression features from neighbours, eventually updating the latent representation of cells and the final reconstructed gene expression profiles. Through evaluation and benchmarking with current available methods, SiGra demonstrates exceptional performance on multiple spatial transcriptomics datasets from different platforms, especially on single-cell spatial profiling. Moreover, the enhanced spatial transcriptomics data by SiGra facilitate insights into cellular communications and underlying biological discoveries.

### SiGra accurately identifies spatial domains in the single-cell spatial profiles of NanoString CosMx SMI
To evaluate the performance of SiGra in deciphering spatial domains, we compare it with five state-of-the-art clustering methods developed specifically for spatial transcriptomics: Seurat v4[10], Scanpy[11], stLearn[12], SpaGCN[14], and BayesSpace[13]. For comparisons, we use the SCST dataset of Lung-9-1 generated by NanoString CosMx SMI. This dataset

**a Graph representation of multimodal single cell spatial transcriptomics**

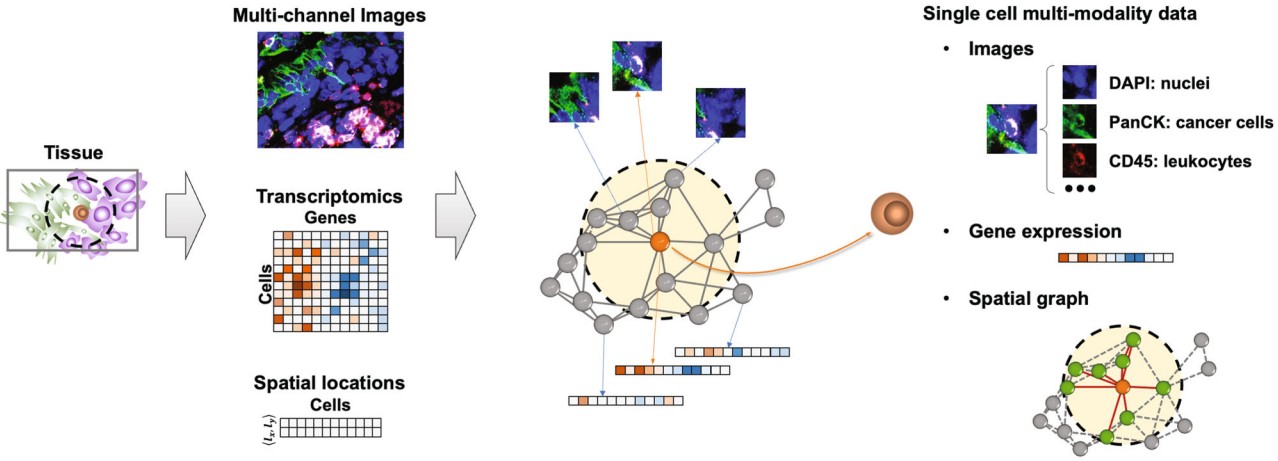

**b Single-cell spatial elucidation through image-augmented graph transformer (SiGra)**

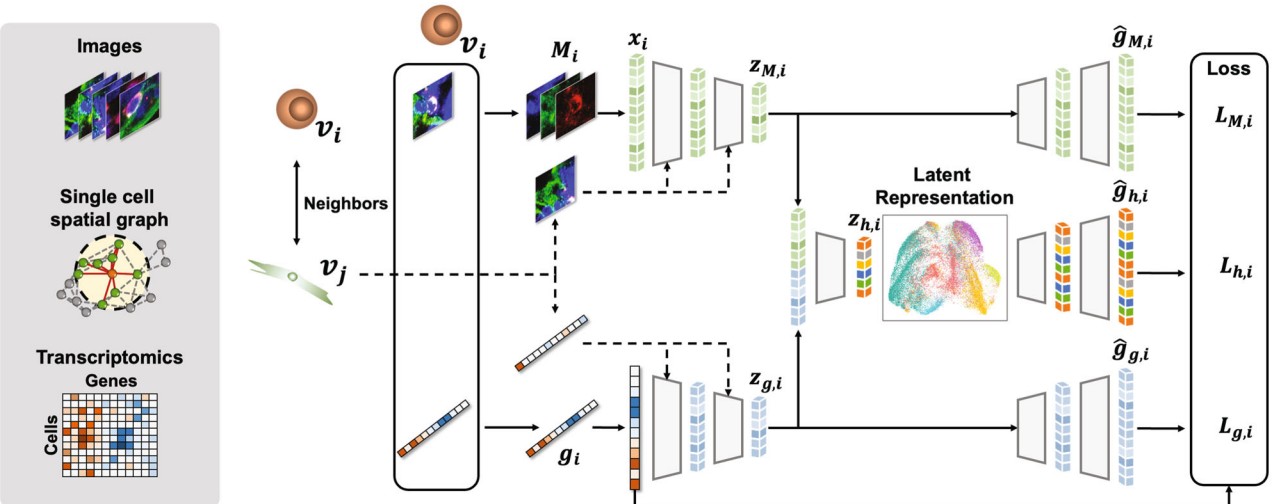

**Fig. 1 | Schematic overview of the SiGra method. a** Graphical representation of the spatial transcriptomics profiles. Each cell on the constructed spatial graph is accompanied by its multichannel images and gene expression. **b** SiGra comprises three graph transformer encoder-decoders (imaging, transcriptomics, and hybrid) with an attention mechanism to incorporate the single-cell multimodal data for simultaneous data enhancement and spatial domain recognition.

consists of 20 FOVs from lung cancer tumour tissue[4], with 982 genes and 83,621 cells that cover eight major cell types, including lymphocytes, neutrophils, mast cells, endothelial cells, fibroblasts, epithelial cells, myeloid cells, and tumours. Details of these experimental data are provided in the Data Availability section. The identified spatial clusters of each method are annotated based on the matched overlap of spatial clusters and ground truth manually annotated in the original study[4].

As described in the Methods, cell types are first identified and then organized into spatial domains. With the spatially heterogeneous cell types identified by different methods, the spatial organization of the 20 FOVs is shown in Fig. 2a. Specifically, SiGra detects the spatial distributions of different cell types that agree well with the original study, i.e., ground truth, with an overall adjusted Rand index (ARI) of 0.55, which is higher than that of Seurat (ARI = 0.37) and BayesSpace (ARI = 0.23). Seurat and BayesSpace significantly mislabelled more cells than SiGra across the 20 FOVs. Meanwhile, the other two methods show much lower accuracy (ARIs: 0.25 for Scanpy, 0.22 for SpaGCN, and 0.34 for stLearn). Of note, the addition of multichannel images (ARI = 0.59) improves the performance by 47.5% compared with using gene expression only as input (median ARI = 0.40). These results

demonstrate that multimodal spatial information contributes to the superior performance of SiGra.

Figure 2b shows the ARI scores of all 20 FOVs. Notably, SiGra is shown to identify the most accurate spatial clusters of different cell types (median ARI = 0.59). Its performance is especially better than that of stLearn (median ARI = 0.22) and Scanpy (mean ARI = 0.25). Compared with SpaGCN (mean ARI = 0.27), Seurat (median ARI = 0.38) and BayesSpace (mean ARI = 0.32), SiGra has relatively better performance with identified clusters more consistent with manual annotations. These comparison results demonstrate that SiGra improves the identification of spatial clusters compared with existing methods for single-cell spatial profiles.

The spatial clustering results are further scrutinized at the individual FOV level (Fig. 2c). Of note, SiGra shows consistency between its identified cellular identities and the ground truth, with the continuous tumour region infiltrated with scattered immune cell clusters. In contrast, BayesSpace and Seurat misidentify the cellular anatomies as either a mixture of fragmental cell regions (FOV-1) or as highly blended cell types (FOV-2). For FOV-1, BayesSpace misidentifies the neutrophil as lymphocytes; meanwhile, it incorrectly identifies some tumour cells as myeloid cells or neutrophils. Seurat fails to disentangle epithelial

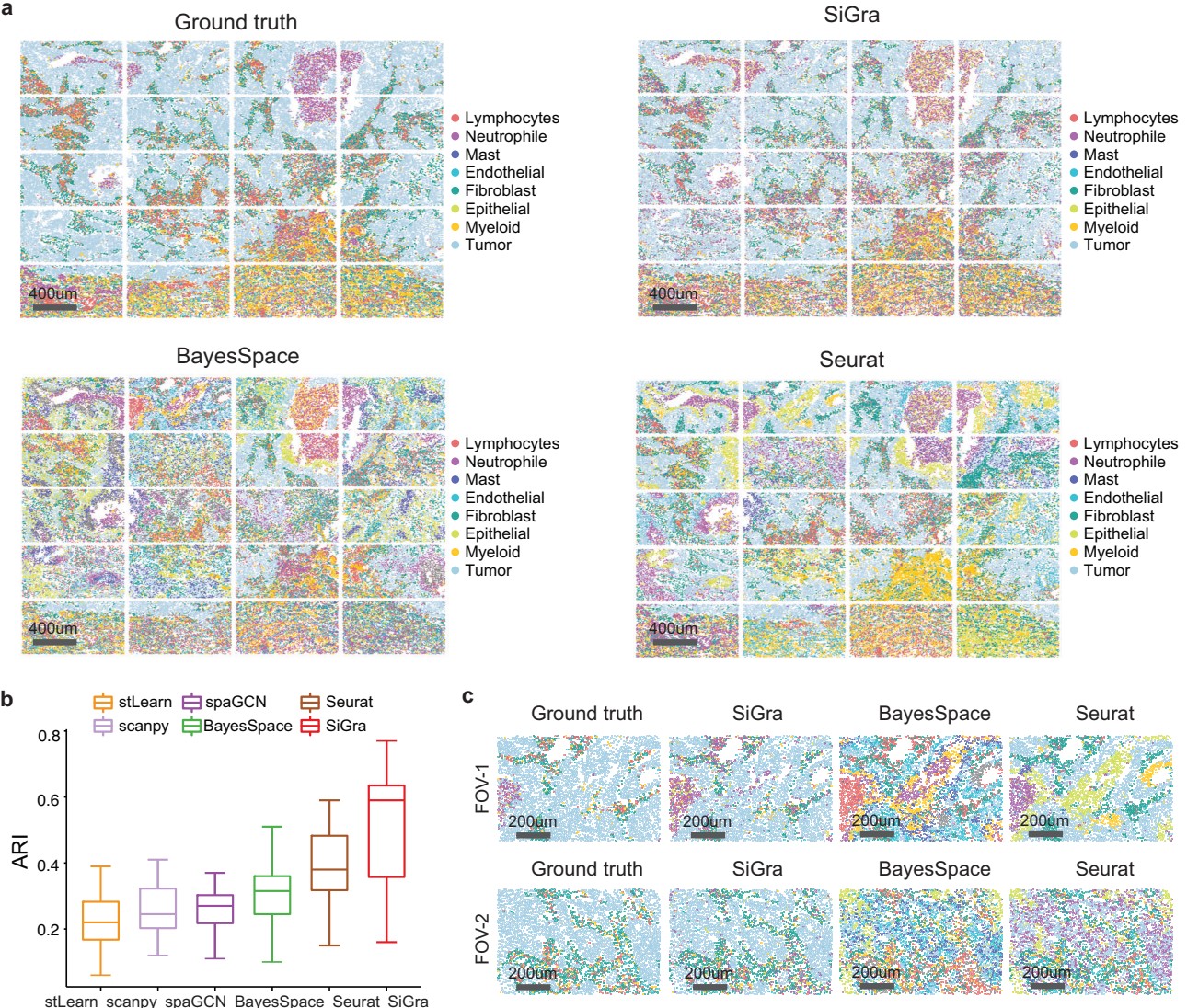

**Fig. 2 | SiGra accurately identifies spatial domains in the single-cell spatial profiles of NanoString CosMx SMI. a** Spatial regions of ground truth and those detected by different methods, including SiGra, BayesSpace, and Seurat. **b** Boxplot of the adjusted Rand index (ARI) scores of six methods in all 20 FOVs of lung cancer tissue. In the boxplot, the centreline, box limits and whiskers denote the median, upper and lower quartiles, and 1.5× interquartile range, respectively. Source data are provided as a Source Data file. **c** Spatial regions of two FOVs detected by different methods, including SiGra, BayesSpace, and Seurat.

cells from tumour cells. In FOV-2, BayesSpace misidentifies fibroblasts as a mixture of myeloid cells and lymphocytes, while Seurat mixes neutrophils with tumour cells without clear dissection of spatial heterogeneity. These results indicate that the compared methods lack the capability of deciphering major spatial regions in SCST data.

The performance of the spatial domains identified by summarizing the detected spatially heterogeneous cell identities is shown in Supplementary Fig. 5. The tumour slice is pathologically annotated as the tumour region (green), the desmoplasia region (red), and the adjacent normal region (orange). SiGra achieves an ARI of 0.60, better than other methods, including BayesSpace (ARI = 0.25), SpaGCN (ARI = 0.10), Seurat (ARI = 0.10), stLearn (ARI = 0.10), and Scanpy (ARI = 0.17). These results show that SiGra identifies reliable spatial domains based on its accurately identified cell identities in SCST data.

### SiGra enhances gene expression patterns that distinguish intratumoral spatial heterogeneity

SiGra enhances the spatial gene expression data and improves downstream analysis for unveiling biological relevance. Herein, we perform UMAP on raw data and enhanced data (Fig. 3a and Supplementary

Fig. 1b). Apparently, the enhanced data reveal better data topology with different cell types better separated in the UMAP results. Moreover, the enhanced cell type-specific gene markers show prevalently consistent expression in their corresponding cell types (Fig. 3b). For example, the enhanced fibroblast marker gene *DCN*[19] demonstrates uniform high expression in fibroblasts and low expression in other cell types. In contrast, in raw data, *DCN* presents sporadic expression in fibroblasts but is highly expressed in other nonfibroblasts. Thus, SiGra not only denoises false-positive expressions (e.g., *DCN* expression in nonfibroblasts) and extreme values but also imputes missing values (e.g., missing values of *DCN* expression in fibroblasts). Meanwhile, SiGra-enhanced data exhibit topological expression of cell type-specific markers (Fig. 3c). For example, after enhancement, *CD68*[20] and *MGP*[21] show elevated expression in myeloid- and endothelial cell-enriched regions, respectively. The tumour-specific genes *EPCAM*[22], *SOX4*[23], and *KRT7*[24] show strong and uniform enrichment in tumour regions, while these genes are not captured in the raw data of some tumour cells (Fig. 3d). In addition, the enhanced SCST data are also more comparable to the bulk RNA-seq data than the raw SCST data (Supplementary Fig. 2a and Supplementary Note 1). These results

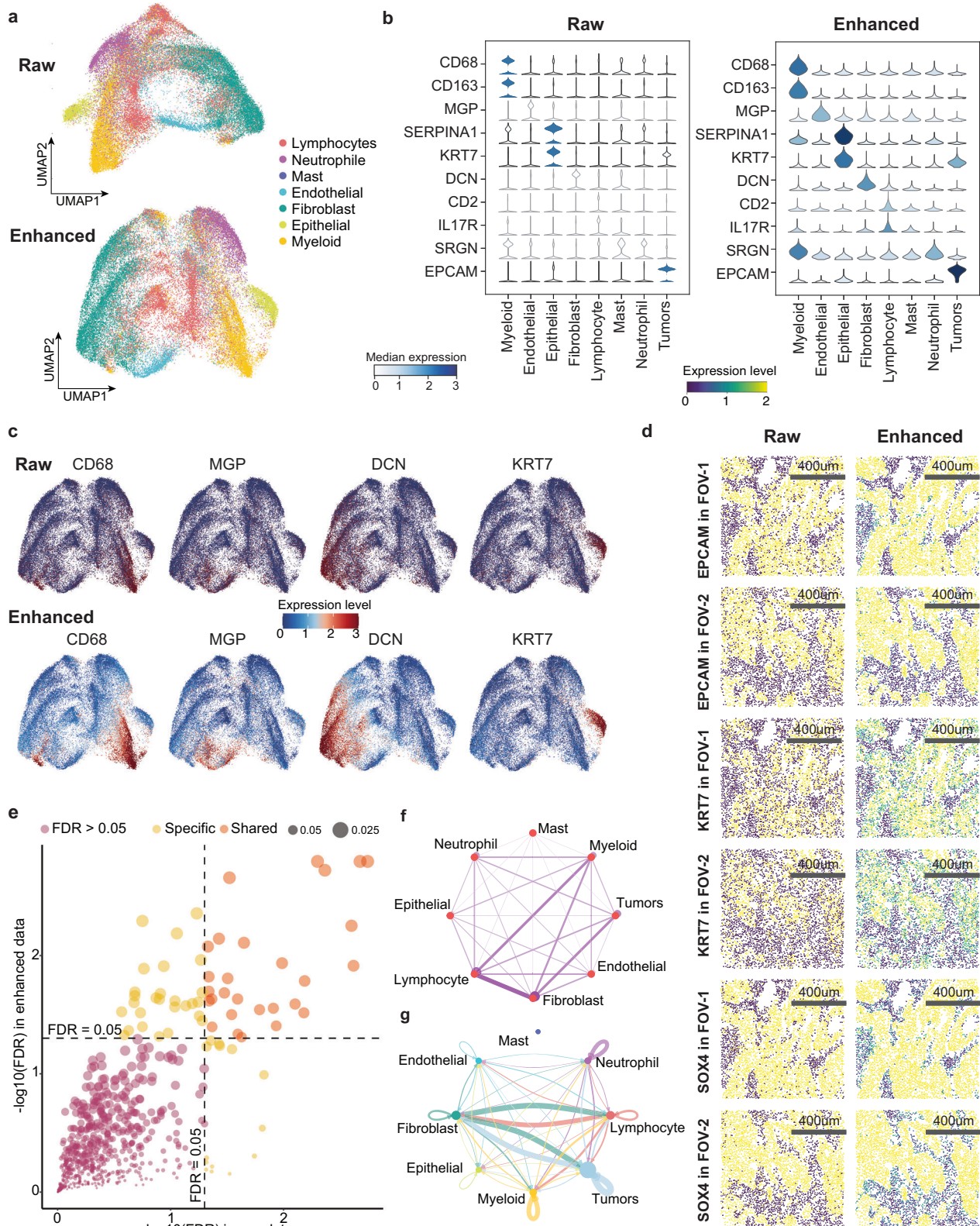

**Fig. 3 | SiGra enhances gene expression patterns that distinguish intratumoral spatial heterogeneity. a** UMAP visualizations of raw data and the enhanced data of the nontumor cell population. **b** Violin plots of the raw expression and the enhanced expression of marker genes in different cell populations (endothelial: 4272; epithelial: 3458; fibroblast: 13,368; lymphocyte: 11,664; mast: 287; M-cell: 7560; neutrophil: 5731; tumors: 37,281). **c** UMAP visualization of the raw expression and the enhanced expression of marker genes (*CD68, MGP, DCN, KRT7*). **d** Spatial visualization of the raw expression and the enhanced expression of tumour-related genes (*EPCAM, KRT7, SOX4*) in two FOVs. **e** Scatter plot of significantly associated ligand–receptor (L-R) pairs in the raw data and the enhanced data. Orange- and yellow-coloured dots represent the shared and specific L-R pairs, respectively. Source data are provided as a Source Data file. **f** Network representation of the weighted adjacent cell types. The edge width represents the adjacency between two cell types. **g** Adjacent cell communications considering both neighbouring cell weights and L-R interaction strength.

demonstrate the capability of SiGra to improve gene expression data for better spatial cellular characterization.

To further prove that the data enhanced by SiGra are useful for downstream analysis, we collect a combined list of candidate ligand–receptor (L-R) pairs from the International Union of Pharmacology (IUPHAR)[25], Connectome[26], FANTOM5[26], HPRD[27], the Human Plasma Membrane Receptome (HPMR)[28], and the Database of Ligand–Receptor Partners (DLRP)[29], which encompass 815 ligands, 780 receptors, and 3,398 reliable L-R interaction pairs. Among them, genes of 660 L-R pairs are included in this NanoString CosMx dataset. Across the 660 total L-R interactions, 55 L-R pairs of enhanced data and 42 L-R pairs of raw data were significantly associated (false discovery rate (FDR) < 0.05), with 28 L-R pairs shared between enhanced and raw data (Fig. 3e). Further interrogation indicated that those raw-specific L-R pairs were more likely to be false positives, which may result from the original data noise and the low data quality (Supplementary Fig. 2b and Supplementary Note 1). These L-R-associated pairs in the enhanced data have great biological relevance and facilitate the mining of cellular communications. Specifically, we further reveal the adjacencies between different cell types (Fig. 3f, Methods) as well as the cell–cell communications considering both the neighbouring cell weights and the L-R communication strength (Fig. 3g). Tumour-associated fibroblasts play a central role in the tumour microenvironment and are not only adjacent to tumour cells and lymphocytes (Fig. 3f) but also demonstrate strong communication with them (Fig. 3g). In contrast, lymphocytes and myeloid cells are close to each other but have less communication. Collectively, this evidence demonstrates that the enhanced data contribute to downstream analysis and are crucial for revealing cellular interactions, which otherwise would be hidden due to data sparsity.

## SiGra enhances the single-cell spatial data of Vizgen MERSCOPE

SiGra is further evaluated on the other SCST dataset of mouse livers compiled by Vizgen MERSCOPE, which consists of 347 genes and 395,215 cells. In this dataset, SiGra reveals different spatial cell clusters (Supplementary Fig. 2c). For better visualization, we focused on the four major cell clusters (Fig. 4a). Cluster 1 (C-1) and cluster 2 (C-2) are located adjacent to the central and portal veins, respectively, while cluster 3 (C-3) and cluster 4 (C-4) are located at blood vessels. Importantly, the enhanced data by SiGra reveal histologically meaningful liver-specific gene expression patterns in different regions (Fig. 4b). For example, SiGra remarkably enhances hepatocyte hallmark genes *Cyp2c38*[30] and *Axin2*[31], which are predominantly expressed near blood vessels. The endothelial cell markers *Cd34*[32] (Fig. 4b) and *Vwf*[33] (Supplementary Fig. 2d) were also clearly present in the central veins, portal veins, and sinusoids. The raw data, in contrast, show noisy expression of these genes in the nonrelevant anatomic regions. For example, *Cd34* and *Vwf* show scattered false signals in the nonblood-vessel regions and missing expression in smaller veins, especially sinusoids. Thus, essential cellular anatomical structures in the liver tissue, such as central veins, portal veins, and sinusoids, can be clearly identified by the enhanced expressions of *Cd34*, *Vwf*, and *Axin2*, but not the noisy raw data, which are further confirmed by the UMAP plots (Fig. 4c). From the boxplots of the expression of these hallmark genes (Fig. 4d), both C-1 and C-2 are suggested to be hepatocytes (high *Cyp2c38* and *Axin2* expression), C-2 is enriched with periportal hepatocytes (higher *Axin2* expression), C-4 contains mainly endothelial cells (high *Cd34* and *Vwf* expression), and C-3 is likely to be hepatic stellate cells. Notably, the enhanced cell-type specific genes are only enriched in their restricted regions but not in irrelevant regions, suggesting that SiGra does not introduce noticeable artefacts in the enhanced data. Further comparison between enhanced data with bulk RNA-seq shows the improved spatial data quality by SiGra (Supplementary Fig. 2e and Supplementary Note 1).

In addition, more differentially expressed genes (DEGs) were identified from the enhanced data than from the raw data (Fig. 4e). For example, the enhanced data recovered 59, 42, and 35 DEGs for C-1, C-2, and C-3, respectively, while the raw data only identified 12, 13, and 12 DEGs, respectively. The identified DEGs in the enhanced data also showed a higher average log2-fold change (logFC; C-1: 0.8; C-2: 0.89; C-3: 0.79) than the raw data. The large overlaps between the DEGs revealed by the enhanced spatial data and the single-cell RNA-seq data further verify the improved data quality after enhancement (Supplementary Fig. 2f and Supplementary Note 1). Moreover, the enhanced data reveal meaningful and associated L-R pairs. As shown in Fig. 4f, among the 64 L-R pairs identified in this dataset, 13 L-R pairs in the enhanced data and 12 L-R pairs in the raw data were significantly associated (FDR < 0.05), with 9 of these L-R pairs shared between the enhanced and raw data. Further investigation revealed that the enhanced-specific L-R pairs also presented strong associations in the bulk RNA-seq data of mouse livers[34] (*Wnt2-Fzd4*: 0.581; *Pkm-Cd44*: 0.885; *Col1a2-Itga2b*: 0.641; *Dll1-Notch2*: 0.798). However, the raw-specific L-R pairs have no associations in the bulk data, indicating that those raw-specific L-R pairs are more likely to be false positives. These results demonstrate that SiGra facilitates the recovery of liver-specific genes and L-R interactions, which enables better characterization of the spatial architecture of mouse liver tissue.

## SiGra improves the identification of known layers in brain tissues

To show that SiGra not only outperforms existing methods in single-cell spatial data but also in spot-based spatial transcriptomics data, here we analyse the 10x Visium datasets from the human dorsolateral prefrontal cortex (DLPFC). These datasets consist of 12 tissue slices of human brains, covering up to six neuronal layers and white matter manually annotated by the original study. To evaluate the benchmarking performance, the identified spatial clusters are annotated based on the matched overlap of spatial clusters and ground truth. Figure 5a shows the ARI scores for all 12 tissue slices (Supplementary Figs. 3 and 4), on which SiGra (median ARI: 0.57) outperforms Scanpy (median ARI: 0.28), Seurat (median ARI: 0.29), stLearn (median ARI: 0.39), SpaGCN (median ARI: 0.40), and BayesSpace (median ARI: 0.44).

We further examined the DLPFC anatomical structures identified by different methods. For tissue slice 151,507 (Fig. 5b), SiGra reveals more accurate spatial regions than the other methods. Seurat identifies Layer 4 scattered in the regions of Layers 3 and 5 without clear boundaries. Scanpy, stLearn, and BayesSpace are not able to distinguish the anatomical shape of Layer 4. Figure 5c shows the other tissue slice 151,676 with spatial regions identified by different methods. Only SiGra deciphers the layer boundaries clearly, reaching good agreement with manual annotations (ARI = 0.62), while other methods can only achieve ARIs of less than 0.4. Specifically, stLearn intermingles Layer 2 with Layer 3, with additional mixtures of Layer 4 and white matter. BayesSpace mixes Layer 4 with Layer 5 and misidentifies some white matter as Layer 2, which leads to its poor performance. Interestingly, although stLearn also utilizes histology information from the haematoxylin and eosin (H&E) images to capture morphological features, its performance is substantially worse than that of SiGra, suggesting that SiGra incorporates multimodal spatial features in a more effective way. In addition, based on the latent embeddings of slice 151676 obtained by different methods (Fig. 5d), SiGra presents much clearer separations of different anatomical layers, while Scanpy and SpaGCN only discern white matter, failing to distinguish other neuronal layers. All these benchmarking results show that SiGra is able to better identify subtle spatial domains than other methods in spot-based spatial transcriptomics data.

## SiGra improves spatial gene expression for better structural characterization

To further validate that SiGra enhances spatial gene expression, we detected the DEGs of each domain in slice 151676. Compared with

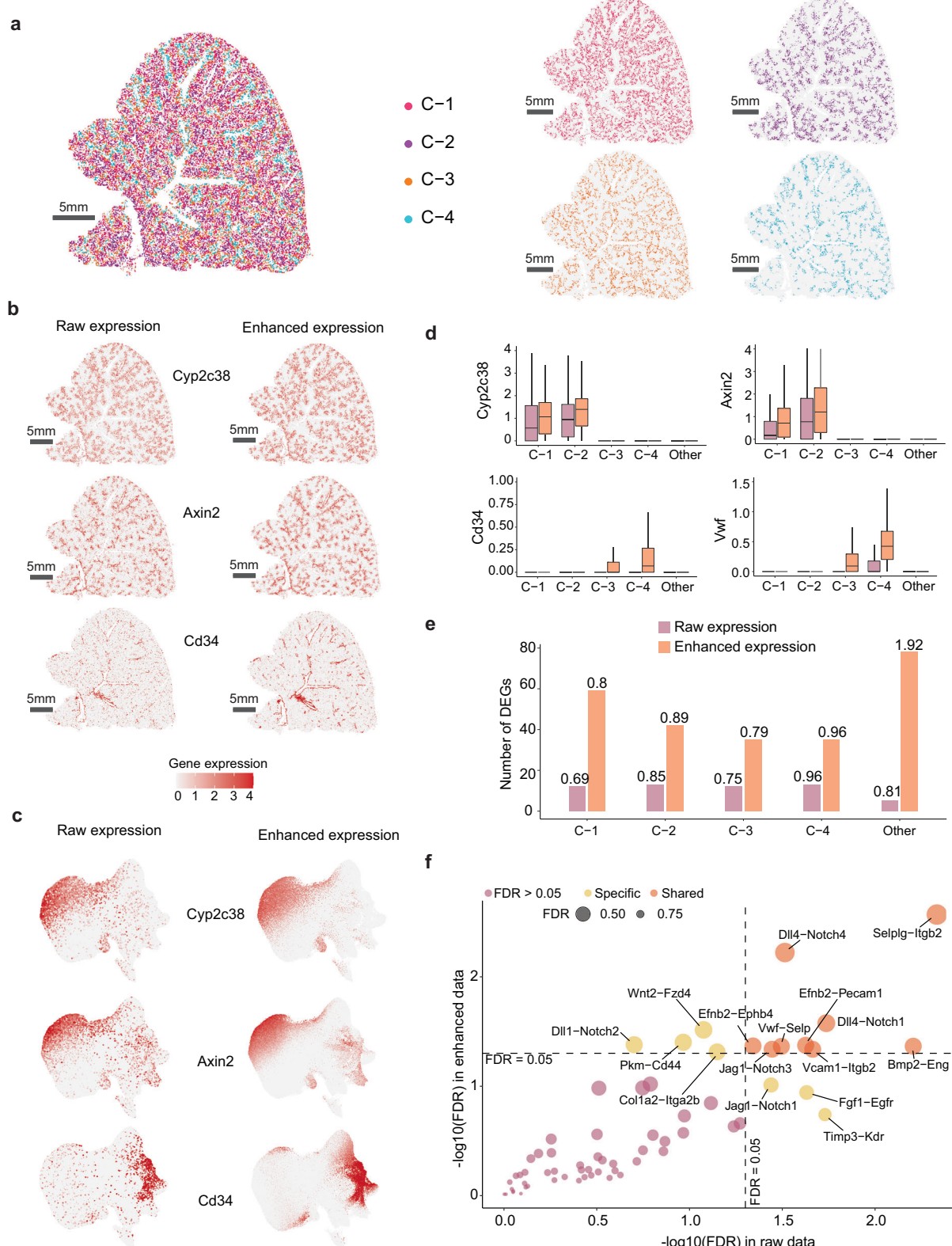

**Fig. 4 | SiGra enhances the single-cell spatial data of Vizgen MERSCOPE.**
**a** Spatial visualization of the cell clusters in the single-cell spatial data from mouse liver tissue. **b** Spatial visualization of the raw expression and the enhanced expression of liver-related genes (*Cyp2c38, Axin2, Cd34*). **c** UMAP visualization of the raw expression and the enhanced expression of liver-related genes (*Cyp2c38, Axin2, Cd34*). **d** Boxplots of the raw expression and the enhanced expression of liver-specific genes in different cell clusters (C1: 63,300; C2: 44,173; C3: 39,823; C4:

29,170; Other: 190,645). In the boxplot, the centreline, box limits and whiskers denote the median, upper and lower quartiles, and 1.5× interquartile range, respectively. **e** Comparisons of the number of differentially expressed genes (DEGs) in cell clusters. The labelled number is the average logFC for that cell cluster. **f** Scatter plot of significantly associated ligand–receptor (L-R) pairs in the raw data and the enhanced data. Orange- and yellow-coloured dots represent the shared and specific L-R pairs, respectively. Source data are provided as a Source Data file.

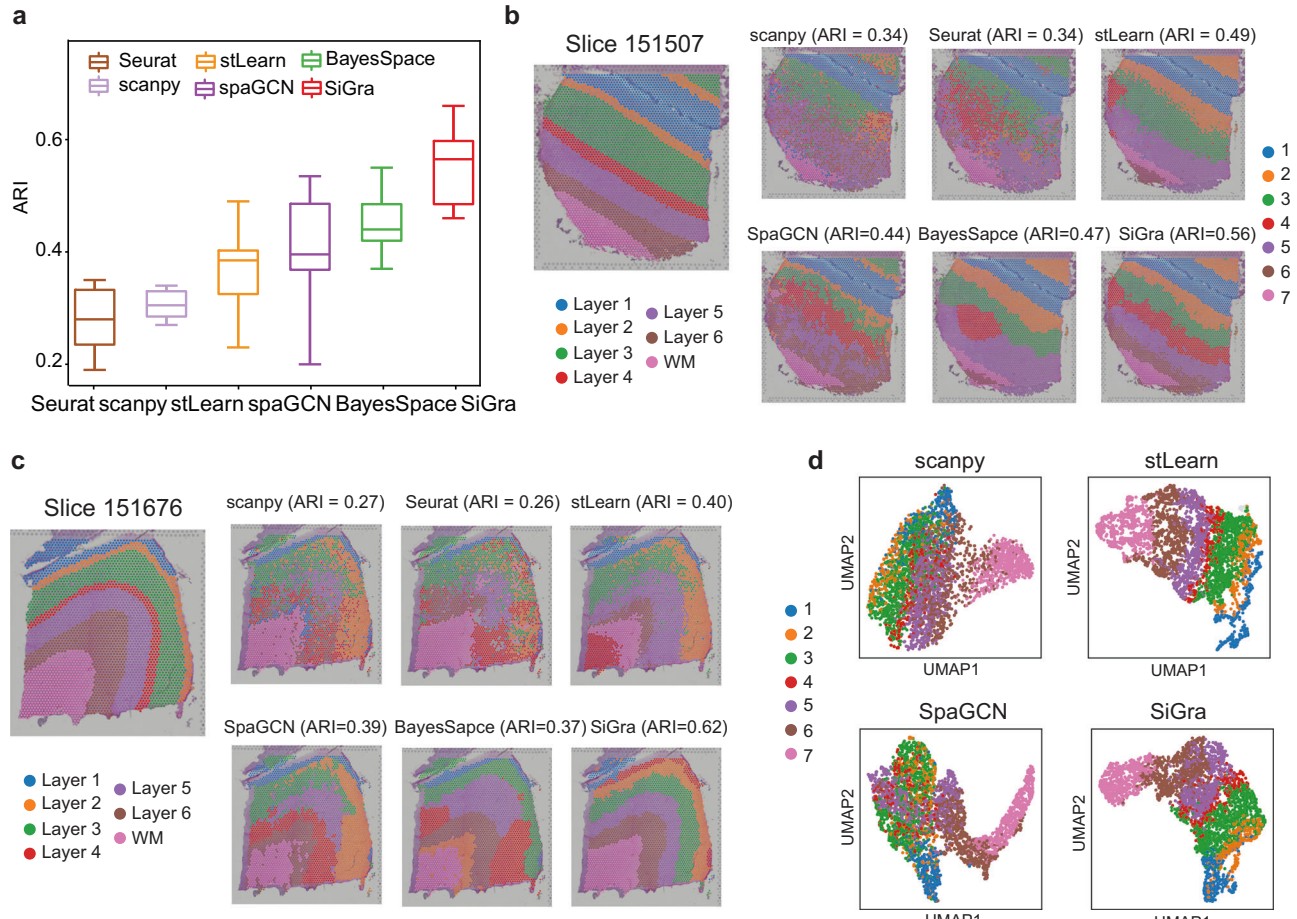

**Fig. 5 | SiGra improves the identification of known layers in brain tissues.** **a** Boxplot of the adjusted Rand index (ARI) scores of six methods in all 12 DLPFC slices. In the boxplot, the centreline, box limits and whiskers denote the median, upper and lower quartiles, and 1.5× interquartile range, respectively. Source data are provided as a Source Data file. **b** Spatial domains detected by different methods, including Scanpy, Seurat, stLearn, BayesSpace, SpaGCN, and SiGra, in DLPFC slice 151507. **c** Spatial domains detected by different methods, including Scanpy, Seurat, stLearn, BayesSpace, SpaGCN, and SiGra, in DLPFC slice 151676. **d** UMAP visualizations of latent embeddings generated by Scanpy, stLearn, SpaGCN, and SiGra in DLPFC slice 151676. SpaGCN and BayesSpace are not shown because they did not provide latent embeddings for UMAP visualization.

the number of DEGs detected in raw data, SiGra detects more DEGs specific to individual regions (Supplementary Data 1). For example, with 232 DEGs of Layer 1 detected in raw data, 595 DEGs are found specific to Layer 1 from enhanced data. Moreover, for each of the neuronal layers, region-specific marker genes can be better identified after SiGra data enhancement (Fig. 6a). For example, *MYH11*[35] presents enriched expression in Layer 1 (logFC = 2.96). *C1QL2*[36] and *CUX2*[36] are overexpressed in Layer 2 and Layer 3, with logFC values of 1.74 and 1.44, respectively. *SYT2*[37] and *FEZF2*[38] are enriched in Layer 4 (logFC = 1.31) and Layer 5 (logFC = 1.42). *PAQR6*[39] shows dominantly enriched expression (logFC = 2.9) in the white matter area. In contrast, these marker genes do not show clear expression patterns in raw data, indicating the limits that raw data face in distinguishing spatial domain boundaries. Violin plots further show the expression of marker genes in raw data and enhanced data (Fig. 6b). Such enhanced gene expression patterns are also observed in other DLPFC slices, for example, slice 151507 (Fig. 6c and Supplementary Data 2), where *RELN*[40] (logFC = 3.24) and *ADCYAP1*[41] (logFC=2.81), i.e., markers of Layer 1 and Layer 2 present remarkable enhancement, in contrast to their sporadic expressions in raw data. In addition, we examined the layer-enriched gene markers identified in the enhanced data, which showed high consistency with those of the original study[42] (Supplementary Note 2 and Supplementary Data 3). These results demonstrate the capability of SiGra to reduce noise

and improve gene expression patterns in spot-based spatial transcriptomics data.

## Discussion

Spatial biology technology has rapidly evolved into the single-cell era[2]. Commercially available in situ hybridization platforms such as Nano-String CosMx SMI and Vizgen MERSCOPE have enabled spatial gene expression profiling at subcellular resolution (50 nm) for 500–1000 targeted genes. Experimental in situ sequencing technologies such as ExSeq[43] expand SCST to the whole transcriptome. Spot-array spatial transcriptomics technologies such as Stereo-seq[44] and Seq-Scope[45] are also reaching subcellular resolution (500 nm–600 nm). However, in all these technologies, the resulting SCST data are limited by the low total transcriptions per cell, noisy data, and substantial zeros, which raises challenges in effective downstream analysis[15]. To accurately reveal spatial and cellular anatomic structures and to enhance noisy gene expression data, we developed the SiGra method, a graph artificial intelligence model, to incorporate multimodal data, including multi-channel IHC images, spatial adjacency cell graphs, and gene expression. The use of graph transformers over a spatially adjacent cell graph as well as the imaging-transcriptomics hybrid architecture allows SiGra to effectively leverage the rich information from the high-content IHC images as well as the spatial distribution. In SiGra, the multimodal information from images and original transcriptomics are summarized

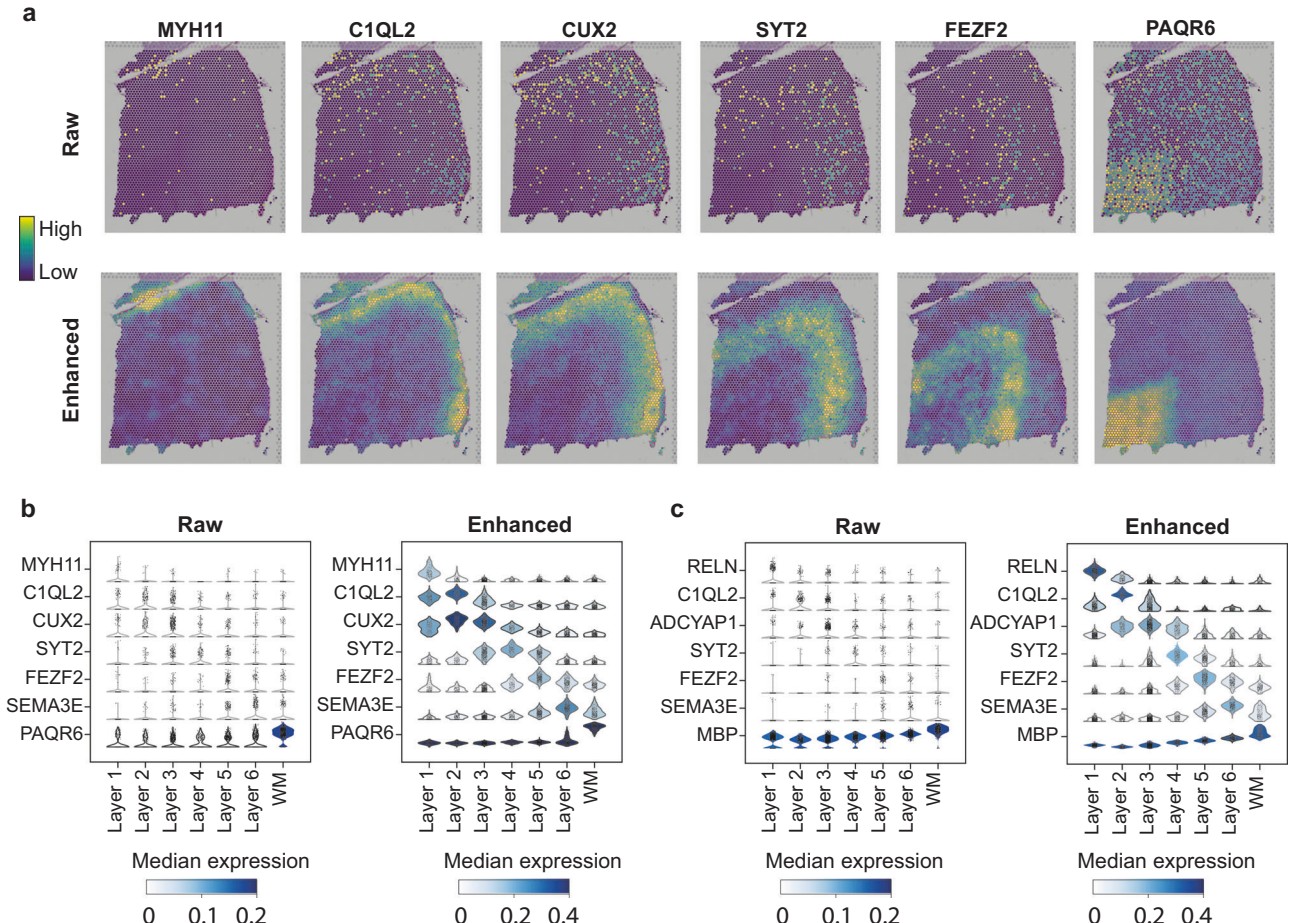

**Fig. 6 | SiGra improves spatial gene expression for better structural characterization. a** Spatial visualization of the raw expression and the enhanced expression of marker genes (*MYH11, C1QL2, CUX2, SYT2, FEZF2, PAQR6*) in DLPFC slice 151676. Numbers of spots in each layer: Layer 1: 289; Layer 2: 254; Layer 3: 836; Layer 4: 254; Layer 5: 649; Layer 6: 616; WM: 533. **b** Violin plots of the raw expression

and the enhanced expression of marker genes in DLPFC slice 151676. **c** Violin plots of the raw expression and the enhanced expression of marker genes in DLPFC slice 151,507. Numbers of spots in each layer: Layer 1: 817; Layer 2: 305; Layer 3: 1,215; Layer 4: 369; Layer 5: 675; Layer 6: 486; WM: 354.

at the single-cell level, with the information from neighbouring cells selectively captured by the attention mechanism. With these technical advances, the SiGra model outperforms existing methods and significantly improves downstream data analysis.

SiGra is designed to utilize multimodalities, including multichannel images of cells and their niches, to address technological limitations and achieve augmented spatial profiles. Designed as a general-purpose tool for spatial transcriptomics data enhancement and spatial pattern profiling, SiGra can be directly used not only for SCST but also for spot-based spatial transcriptomics data. SiGra demonstrates superior performance on three different platforms, in both healthy and diseased tissues, and across different species, which provides a general solution for existing spatial transcriptomics data analysis pipelines. Notably, SiGra can identify spatial domains at different resolutions, depending on the data types and the applications. For the spot-level spatial data that have a low spatial resolution and consist of mixed cells/cell types in each spot, SiGra directly and accurately identifies the spatial structures (Fig. 5), such as the anatomical layers in the brain cortex on DLPFC slices, by clustering the latently represented spots. For the SCST with significantly high resolution, SiGra identifies spatial regions at the cellular level (Fig. 2). Meanwhile, on such high-resolution SCST data, SiGra is also able to reveal the regional anatomic spatial structures by further summarizing the Leiden clustering results with a dimensional moving window approach (Supplementary Fig. 5 and Supplementary Note 3). The

effects of autofluorescence on SiGra are also examined in Supplementary Fig. 6 and Supplementary Note 4.

In addition to its superior performance and technical advantages, SiGra can be further improved in the future. First, as newer spatial omics technologies[46] continue evolving and new data modalities continue to emerge, SiGra can be improved by incorporating new omics data types, new image types, 3-D spatial information, etc., to extend data exploration. The hybrid architecture allows SiGra to adapt additional spatial information and incorporate multiomics data. As an advanced deep learning model, SiGra also faces the limitations of the black-box nature of artificial intelligence[47–49]. This can be ameliorated through downstream analysis, such as cell–cell interaction analysis. Further development of SiGra will enhance the model interpretability that can address some of the problems and provide insights into the underlying mechanism in tissue ecosystems. As the capacity and efficiency of experimental technologies continue to improve, SiGra is anticipated to facilitate biological discoveries and insights into complex tissues and diseases.

## Methods
### Data preprocessing and graph representation
Spatial transcriptomics data generated by different platforms, including the NanoString CosMx™ SMI lung cancer dataset (Lung-9-1)[4], Vizgen MERSCOPE mouse liver dataset L1R1 released in January 2022[15], and 10x Visium datasets from the human dorsolateral

prefrontal cortex (DLPFC)[42], are preprocessed and represented in uniform format (Fig. 1) for SiGra. The original spatial profiles are converted to single-cell (or spot) images, single-cell (or spot) expression, and a spatial graph of adjacent cells (or spots), which serve as the input for SiGra.

Regarding the NanoString Lung-9-1 dataset, the composite images of the DAPI, PanCK, CD45, and CD3 channels from 20 FOVs, the cell centre coordinates (from the cell metadata file), and the single-cell gene expression file of 960 genes are used. For each cell, four 120-by-120 pixel (21.6 μm-by-21.6 μm) images with the cell at the centre are cropped from the images. Edges in the spatial adjacency graph are constructed if the cell-to-cell distance (Euclidian distance) ≤ 80 pixels (14.4 μm). NanoString's annotations of cell types are obtained from their provided Giotto object. Regarding the Vizgen L1R1 dataset, images of the DAPI staining and the three IHC boundary staining, the single-cell expression data of 347 genes, and the cell centre coordinates are used. The images in the middle of the z-packs (z3) are used, as recommended by Vizgen. These images are cropped into single-cell 200-by-200 pixel (21.6 μm -by-21.6 μm) images. Edges in the spatial adjacency graph are constructed if the cell-to-cell distance ≤ 150 pixels (16.2 μm). Regarding the 10x Visium DLPFC dataset, the high-resolution H&E images as well as the .h5 files ("filtered_feature_bc_matrix.h5") are used as input. For each spot, three spot-specific images (for the RGB channels) are extracted, with 50-by-50 pixel (38.7 μm-by-38.7 μm) images. The cut-off distance for generating the spatial graph between spots is 150 pixels (116 μm). The top 3000 highly variable genes were identified using the Seurat standard pipeline[10] and used for analysis. For all datasets, the raw counts of gene expression were normalized by multiplying by 10,000, followed by log-transformation. The parameters of the size of single-cell images and the cut-off of cell-to-cell distance for constructing the spatial graph are determined empirically depending on the cellular anatomy of the tissue. In the spatial adjacency graph, most cells have 5–6 neighbouring cells.

In this way, the final graph representation of the original single-cell or spot spatial transcriptomics data is a spatial graph $G = (V, E)$, with $v_i \in V$ representing the $i$ th cell with $i = 1, \cdots, n$ representing the total $N$ cells, $e_{ij} \in E$ representing the spatial proximity between two cells $v_i$ and $v_j$, and $A$ as the adjacency matrix of the graph. Each cell $v_i$ on the spatial graph is accompanied by multichannel images $\boldsymbol{M}_i = \{M_{i,c}\}$, with $c = 1, \cdots, C$ representing each imaging channel, and gene expression $\boldsymbol{g}_i = \{g_{i,k}\}$, with $k = 1, \cdots, K$ representing genes.

## The SiGra model

SiGra is a hybrid multimodal graph transformer framework with three transcriptomics reconstruction modules: the imaging-based encoder-decoder, the transcriptomics-based encoder-decoder, and the hybrid encoder-decoder.

1. Imaging-based encoder-decoder. For a cell $v_i$, the multichannel images $\boldsymbol{M}_i$ are transformed to a vector $\boldsymbol{x}_i \equiv [\text{vec}(M_{i,1}), \cdots, \text{vec}(M_{i,c})]^T$. An encoder with a series of multihead graph transformer layers is used to project the imaging vector to the latent space as $\boldsymbol{z}_{M,i}$ and then reconstruct the gene expression profile of this cell $v_i$ as $\hat{\boldsymbol{g}}_{M,i}$. The images $\boldsymbol{M}_j$ from neighbouring cell $v_j \in \mathcal{N}(v_i)$ are also used as the input, where $\mathcal{N}(\cdot)$ represents the neighbours in the graph $G$.
2. Transcriptomics-based encoder-decoder. The original gene expression profile $\boldsymbol{g}_i$ for cell $v_i$, with $\boldsymbol{g}_j$ for adjacent cells $v_j \in \mathcal{N}(v_i)$ also as the input, is projected to the latent space as $\boldsymbol{z}_{g,i}$, which is then used to reconstruct the gene expression of cell $v_i$ as $\hat{\boldsymbol{g}}_{g,i}$.
3. Hybrid encoder-decoder. The latent representation of the imaging and transcriptomics features are catenated as $[\boldsymbol{z}_{M,i}, \boldsymbol{z}_{g,i}]$, further projected to hybrid latent feature $\boldsymbol{z}_{h,i}$, and then used to reconstruct the gene expression for cell $v_i$ as $\hat{\boldsymbol{g}}_{h,i}$ through graph

transformer layers. The latent features $\boldsymbol{z}_{M,j}$ and $\boldsymbol{z}_{g,j}$ from neighbouring cells $\{v_j\}$ are also used by the graph transformers.

**Graph transformer convolutional layer.** Multihead graph transformer[50] layers with an attention mechanism (Supplementary Fig. 1) are the main components of the SiGra model. Briefly, for a cell $v_i$, the propagation of the graph transformer from the $l$ layer to the $l+1$ layer is defined as $\boldsymbol{h}_i^{(l+1)} = \text{ReLU}(W_1^{(l)} \boldsymbol{h}_i^{(l)} + \sum_{v_j \in \mathcal{N}(v_i)} \alpha_{i,j} V_j^{(l)})$, where the rectified linear unit (ReLU)[51] is used as the nonlinear gated activation function. The attention module is defined as $\alpha_{i,j} = \text{softmax}\left(\frac{\langle Q_i^{(l)}, K_j^{(l)} \rangle}{\sum_{u \in \mathcal{N}(i)} \langle Q_i^{(l)}, K_u^{(l)} \rangle}\right)$, where:

$$\text{query}: Q_i^{(l)} = W_Q^{(l)} \boldsymbol{h}_i^{(l)} + b_Q^{(l)} \tag{1}$$

$$\text{key}: K_j^{(l)} = W_K^{(l)} \boldsymbol{h}_j^{(l)} + b_K^{(l)} \tag{2}$$

$$\text{value}: V_j^{(l)} = W_V^{(l)} \boldsymbol{h}_j^{(l)} + b_V^{(l)} \tag{3}$$

and $\langle Q, K \rangle \equiv \exp\left(Q^T K / \sqrt{\dim(\boldsymbol{h}_i^{(l)})}\right)$. The multihead attention values, whose notation is omitted for simplicity, are concatenated.

**Loss function.** SiGra learns the reconstructed gene expression via a self-supervised loss of combined MSE from gene embedding, image embedding and combined embedding, with the loss function

$$L = \sum_{i=1}^{N} \lambda_1 L_{M,i} + \lambda_2 L_{g,i} + L_{h,i} \tag{4}$$

where $L_{M,i} = \frac{1}{N} \sum_{i=1}^{N} \|\boldsymbol{g}_i - \hat{\boldsymbol{g}}_{M,i}\|_2$, $L_{g,i} = \frac{1}{N} \sum_{i=1}^{N} \|\boldsymbol{g}_i - \hat{\boldsymbol{g}}_{g,i}\|_2$, $L_{h,i} = \frac{1}{N} \sum_{i=1}^{N} \|\boldsymbol{g}_i - \hat{\boldsymbol{g}}_{h,i}\|_2$. In this work, the optimal parameters of $\lambda_1$ and $\lambda_2$ were determined through grid-based hyperparameter fine turning. For single-cell spatial transcriptomics data, the optimal parameters are $\lambda_1 = 0.1$ and $\lambda_2 = 0.1$. For 10x Visium data, the optimal parameters are $\lambda_1 = 1$ and $\lambda_2 = 1$. Details are provided in Supplementary Fig. 7a and Supplementary Note 5.

The other hyperparameters of SiGra include two graph transformer layers for the imaging and the transcriptomics encoders (with dimensions of 512 and 30 for the 1st and the 2nd layers, respectively), one graph transformer layer for the hybrid encoder (based on gene embeddings and image embeddings from the transcriptomics encoder and from the imaging encoder), and two graph transformer layers for imaging, transcriptomics, and hybrid decoders (where the dimension of the first layer is 512, and the dimension of the second layer is same as that of the corresponding transcriptomics data). These hyperparameters were determined by grid search (Supplementary Fig. 7b and Supplementary Note 5). After training, SiGra outputs the hybrid reconstruction $\hat{\boldsymbol{g}} = \{\hat{\boldsymbol{g}}_{h,i}\}$ as the final enhanced expression profile. The latent representation, $\boldsymbol{z} = \{\boldsymbol{z}_{h,i}\}$, of the original SCST data is used for spatial data clustering with the Leiden algorithm[52] from the SCANPY package[11].

## Spatial domain detection

For the spot-level spatial data that have a low spatial resolution and consist of mixed cells/cell types in each spot, SiGra directly detects the spatial domains by clustering the latent-represented spots using Leiden. For single-cell spatial data, SiGra first identifies the cell types for each individual cell by clustering the latent representation using Leiden and then reveals spatial domains via a dimensional moving window agglomeration approach[53]. Specifically, the spatially distributed cells are summarized by a circular window of diameter $d$ sliding in both the $x$ and $y$ directions across the whole image with a given stride length $s$. At each stop $C_{i,j}$ with the coordinate $(x_i, y_j)$, a vector $c_{i,j} \equiv [q_1, \ldots, q_t]$

representing the proportions of the SiGra-identified clusters ($t$) covered by the sliding window is calculated. All the stops $\{C_{i,j}\}$ are recursively merged to $k$ groups $\{a_1,\ldots,a_k\}$ by hierarchical clustering according to the cluster proportion vectors $\{c_{i,j}\}$. These agglomerated groups are defined as spatial domains. The window radius $d$ used in Supplementary Fig. 5 is 100 μm, which is consistent with the 10x Visium spatial resolution, with a stride $s$ of 10 μm. For fair comparisons, the same moving window agglomeration approach is used in benchmarking methods (Supplementary Fig. 5). The ground truth of the anatomic spatial domains for the DLPFC slices and lung cancer slices was obtained from the original study[7] and the certified pathologist at Indiana University Health (T.H.).

### Benchmarking methods and comparison measurement

To evaluate the performance of SiGra, we compare it with five existing methods: Seurat v4[10], Scanpy[11], stLearn[12], SpaGCN[14], and BayesSpace[13]. Seurat and Scanpy are implemented based on their provided vignettes. Briefly, for data preprocessing, 3000 highly variable genes were selected for log normalization, and the top 30 principal components (PCs) were calculated for spatial data clustering. BayesSpace is implemented based on their package vignette. Specifically, the input is the top 15 PCs of the log-normalized expression of the top 2000 HVGs. The nrep parameter is set to 50,000, and the gamma parameter is set to 3. For stLearn, based on its tutorial, the stLearn.SME.SME_normalized() function is performed on raw counts with parameters use_data = "raw" and weights = "physical_distance". The top 30 PCs of the SME normalized matrix are then used for spatial data clustering and visualization. SpaGCN is applied according to its recommended parameters in the package vignette. That is, the top 15 PCs of the log-normalized expression of the top 3000 spatial variable genes are used for spatial data clustering. Two hundred epochs are used for identifying and refining spatial domains. The resolution parameter is selected to ensure that the number of clusters is equal to the ground truth.

Moreover, we comprehensively compared SiGra with MUSE[54] and STAGATE[55] based on simulation data, SCST data, and 10x Visium data. Additional ablation studies were also performed to investigate the contributions of different components in the SiGra model. Details of benchmarking and ablation studies are provided in Supplementary Figs. 8, 9, and Supplementary Note 6.

To evaluate the performance of each method, we use the adjusted Rand index (ARI) to assess the agreement between the identified spatial clusters and the manual annotation. Suppose $\hat{Y}=\{\hat{y}_i\}_{i=1}^n$ represents the spatial clusters and $Y=\{y_i\}_{i=1}^n$ represents the ground truth of $n$ cells divided into $k$ clusters. Then,

$$ARI = \frac{\sum_{ls}\binom{n_{ls}}{2} - \frac{\sum_l\binom{n_l}{2}\sum_s\binom{n_s}{2}}{\binom{n}{2}}}{\frac{\sum_l\binom{n_l}{2}+\sum_s\binom{n_s}{2}}{2} - \frac{\sum_l\binom{n_l}{2}\sum_s\binom{n_s}{2}}{\binom{n}{2}}} \tag{5}$$

where $l$ and $s$ denote the $k$ clusters, $n_l=\sum_i^n I(\hat{y}_i=l)$, $n_s=\sum_i^n I(y_i=s)$, $n_{ls}=\sum_{i,j}^n I(\hat{y}_i=l)I(y_i=s)$, and $I(x=y)=1$ when $x=y$, else $I(x=y)=0$. The ARI ranges from 0 to 1 for increasing match between the identified clusters with ground truth.

### Identifying differentially expressed genes and adjacent cell communications

To identify the differentially expressed genes (DEGs), the Wilcoxon test from the Scanpy package[11] was used. DEGs of each spatial region were selected with a 5% FDR threshold (Benjamin-Hochberg adjustment) and a log2-fold change greater than 1 (logFC > 1).

To reveal the neighbours of each cell type, we aggregate its neighbouring cells by cell type and divide the average number to reveal its weighted neighbours. These weighted neighbours are further used to characterize the adjacent cell communications. Specifically, the interaction strength of each L-R pair is calculated by multiplying their association score and their average expression. Then, we aggregate the interaction strength of each L-R pair by cell type to be the communication strength of two cell types. The neighbouring weight of two cell types is further multiplied by the communication strength of these two cell types for the final adjacent cell communications. In this way, the higher the value of the adjacent communications, the stronger the interaction between two neighbouring cell types.

### Reporting summary

Further information on research design is available in the Nature Portfolio Reporting Summary linked to this article.

## Data availability

The single-cell spatial dataset of NanoString CosMx SMI contains 20 FOVs, which is profiled by the CosMx SMI on Formalin-Fixed Paraffin-Embedded (FFPE) samples of the non-small-cell lung cancer (NSCLC) tissue[4]. The dataset (Lung-9-1) and other lung cancer slices are available from https://nanostring.com/products/cosmx-spatial-molecular-imager/nsclc-ffpe-dataset/. We used the Vizgen MERFISH Mouse Liver Map dataset that contains a MERFISH measurement of a 347 gene panel. Sample L1R1 (liver 1, replicate 1) was used and downloaded from https://info.vizgen.com/mouse-liver-data?submissionGuid=da03b470-e111-425a-b6d2-16d34342f4fe, which includes the list of detected transcripts, gene counts per cell matrix, additional spatial cell metadata, cell boundary polygons, and DAPI images. The human dorsolateral prefrontal cortex (DLPFC) 10x Genomics Visium datasets consists of 12 samples[42]. Each of the samples is manually annotated with up to six cortical layers and white matter. Transcriptomics data and hematoxylin and eosin (H&E) images of corresponding tissue sections are downloaded from http://research.libd.org/spatialLIBD/. Source data are provided with this paper.

## Code availability

SiGra is provided as a Python package available at https://github.com/QSong-github/SiGra[56], with detailed tutorials for general applicability on different SCST platforms. The web interface (SiGra Viewer) is available at http://sigra.sulab.io and enables users to explore the enhanced data in UMAP figures and spatial domains.

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

## Acknowledgements

Q.S. is supported by the Bioinformatics Shared Resources under the NCI Cancer Center Support Grant to the Comprehensive Cancer Center of Wake Forest University Health Sciences (P30CA012197), and the National Institute of General Medical Sciences of the National Institutes of Health (R35GM151089). J.S. is partially financially supported by the Indiana University Precision Health Initiative and by the Indiana University Melvin and Bren Simon Comprehensive Cancer Center Support Grant from the National Cancer Institute (P30CA 082709). J.S. is also supported by R01LM013771.

## Author contributions

Q.S., J.S., and B.Y. developed the structure and arguments and wrote the manuscript. Q.S. and Z.T. implemented the method, applications, and visualization. Z.L. implemented the web interface. T.H. annotated the pathological spatial domains. T.Z. reviewed and edited the manuscript. All the authors reviewed and approved the final manuscript.

## Competing interests

The authors declare no competing interests.
