## [Peer Review File · Nature Communications]

REVIEWER COMMENTS

Reviewer #1 (Remarks to the Author):

This manuscript describes a new method, SiGra, for spatial domain detection and gene expression enhancement for single-cell spatial transcriptomics data. SiGra utilizes three graph transformer autoencoders to integrate the gene expression with image features, then performs spatial clustering and gene expression imputation using the joint embeddings. The paper is clearly written, and I appreciate the large amount of work devoted to this manuscript. However, I am concerned with the ad hoc nature of the method and its effectiveness in practice. In the spatial domain detection task, there is no theoretical justifications for almost all steps of the proposed model. In addition, the model in SiGra is very similar to a published method, MUSE, (<https://www.nature.com/articles/s41587-022-01251-z>), for the same purpose. In the gene expression enhancement task, SiGra does not produce any error control guarantee, either in the form of false discovery control or family-wise error rate control, making it hard to use in practice. No simulations were provided to help build up intuition and understand the performance of SiGra in different settings. My main concerns are:

1. The proposed method is very ad hoc in nature, with no justifications provided for many key steps of the algorithm. The main components of the SiGra are three autoencoders to extract image and gene expression embeddings. Why do we need separate imaging-based and transcriptomics-based autoencoders since the hybrid autoencoder already contains both gene expression and image information? The three autoencoders take data from different modalities as input, but why all of them are used to reconstruct gene expression? How does the attention mechanism balanced outputs from the three autoencoders? Is the final result dominated by embeddings from one autoencoder? The final loss function has three components and two hyper-parameters, λ_1 and λ_2 , are used to weight the gene loss and image loss. Why λ_1 and λ_2 are both set to 0.1? Does that mean the final loss is dominated by the hybrid loss only? Ablation study is needed to investigate the contribution of different components to the model.
2. There are so many parameters in the model, how do the authors determine their optimal choices? The manuscript described the hyper-parameters from line 355 to 362 but did not explain how the optimal set was eventually chosen. Did the authors pick up one that gives the best results? If so, this clearly would lead to model over-fitting and overly optimistic results.
3. The proposed model in this paper is very similar to a recently published method, MUSE (<https://www.nature.com/articles/s41587-022-01251-z>). MUSE also used multiple autoencoders to integrate the gene expression and image information for spatial domain detection, and they performed a detailed ablation study to explain the contribution of each component in their model. The authors should cite and compare with MUSE.
4. The evaluation of spatial domain detection is inappropriate. Spatial domains are regions that show coherence in gene expression and biological function. A spatial domain may contain cells from multiple cell types. In Figure 2a, the authors performed spatial domain detection using SiGra and other methods, then compared the predicted spatial domains with annotated cell types using ARI. This is incorrect since the goal of SiGra and other comparing methods is domain detection, not cell-type identification.
5. Another main application of SiGra is to enhance gene expression for imaging-based ST data. The authors stated that the rationale of this application is such type of data suffers from "missing values, data sparsity, low coverage, and noises"(line 50). However, I cannot agree with this statement. Although the processed gene expression data from CosMx and MERSCOPE are presented as a cell by gene matrix, the raw gene expression is measured using fluorescence imaging with subcellar resolution. The raw data is already very accurate and is in super high resolution. Without further evidence, I do not think there is any need for gene expression enhancement for such type of data.
6. Figure 3 shows many ligand-receptor pairs with low correlations in the raw data have much stronger correlations in SiGra enhanced data. The authors wanted to use this as evidence to

support the usefulness of the enhanced data by SiGra. In my experience, many L-R pairs are loosely correlated due to their relatively low expression level. I doubt whether the strong correlations observed in the enhanced data are true biological interactions or just false discoveries. The authors did not provide any solid evidence to support their findings.

7. Following my above concern, the analysis of the differentially expressed gene using the enhanced data suffers from the same problem

8. No simulations were performed in the study. Simulations would be very helpful in understanding some basic properties of SiGra and its performance in different settings as compared to the other methods.

Reviewer #2 (Remarks to the Author):

(Please see attached document.)

The authors developed Graph Neural Networks based algorithm for spatial domains recovery and transcriptomics data denoising. The authors applied the proposed algorithm to several real datasets and showed some promising results. This is an very interesting work, but there several important issues that need to be addressed. These can be summarized as follows:

Major concern:

\begin{itemize}

\item The terminology of \textit{graph transformer convolutional layer} is misleading. Multi-head attention within the neighborhood has been widely used in the graph neural networks research area. This kind of approach is called "Graph Attention Networks" or "GAT", instead of "Graph Transformer" or " Graph Transformer Convolutional Layer". The crucial difference between these two lies in the way to model graph structural information. "GAT" is a very common and popular graph neural network model. It propagates the node features to the first-order neighbors via attention and gradually extends to high-order neighbors by stacking the graph attentional layers, which is similar to other GNNs, e.g. GCN, GraphSAGE. In contrast, "Graph Transformer" is another family of models that densely connects all the nodes in the graph and use positional encoding or graph neural networks to encode structural information.

\item For the image layer, the name 'autoencoder' is also misleading. The input here is pixel vector and the output is gene expression not the image itself.

\item Still for the image layer, the authors vectorize the images and concatenate them as input. As we know, learning shapes or other topological characters from vectorizing the images is suboptimal. Can the authors provides some insights to support their choice? Also, for each cell, fixed sized square images are extracted as input. Since different cells vary in their sizes, which means many images contain a significant portion of background even info from neighboring cells. I am curious why those extra (noisy) part of image information did not significant affect the performance of the model? More importantly, I am curious the contribution of image to the model performance some ablation studies to illustrate the contribution of image and spatial graph.

\item In Fig2, the authors used the cell label produced by He et al. as ground truth. However, according the the paper, \textit{Cell type was determined by comparing individual cells' expression profiles to reference profiles for different cell types (scRNA-seq and bulk RNA-seq of flow-sorted blood and stroma databases), assigning each cell to the cell type under which its profile was most likely. The likelihood was defined using a negative binomial distribution, with mean defined by a cell type's reference profile plus expected background, and with a size parameter set at 10 to allow for extensive overdispersion.}. Namely, they used an likelihood based method to general cell labels. It is not ideal to treat those label as ground truth. Also, there are 8 slides in the CosMX website, can the author also include the result for the other 7 slides.

\item I also notice that the authors did not include STAGATE for performance comparison (Deciphering spatial domains from spatially resolved transcriptomics with adaptive graph attention auto-encoder.) Can the authors also include STAGATE?

\item For the L-R results in both Figures 3 and 4, can the authors also include some results to evaluate the potential false positive correlations generated by the algorithm using randomly selected two genes? Since SiGra integrates information from neighboring cells, I suspect it will also

create some false correlation results.

`\end{itemize}`

Minor concerns:

`\begin{itemize}`

`\item` For Fig3 A, why the tumor cells are not included?

`\item` In Fig 4, the authors claim 'For example, `\textit{Cd34}` and `\textit{Vwf}` show scattered false signal `\ldots`'. Do the authors have any evidence to support this claim that the expression of `\textit{Cd34}` and `\textit{Vwf}` on those cells are false?

`\end{itemize}`

This is latex code. I have also attached a pdf version.

Reviewer #3 (Remarks to the Author):

I had the chance to review an interesting manuscript that describes a new technique for the improved assessment of spatial transcriptomic data. Single-cell spatial elucidation through image-augmented GRaph transformer (SiGra) incorporates multi-modal spatial features in a more effective manner – using graph transformer autoencoders with attention mechanism – and improves the limits that raw data faces to decipher spatial domains. SiGra is reportedly one of the first methods to utilize multi-channel images in a highly effective manner (true?).

SiGra is benchmarked and compared to other methods of analysis, namely Seurat v4, Scanpy, stLearn, SpaGCN, and BayesSpaces based on previously published datasets published using NanoString CosMx SMI, MERSCOPE, and 10 X Visium.

While the study appears for the most part rigorously executed and analyzed - personally, I am convinced that it is worth trying SiGra – at this point most of the results represent rather indirect evidence that SiGra is indeed superior in unraveling new and important biological mechanisms.

More direct evidence including some experimental validation for their claims is required to justify publication.

1) The results in the DLPFC in this study must be carefully compared to Maynard et al., 2021 – who used gene-level statistics from both layer-specific versus cell type-specific expression profiles for “spatial registration” in their study. What about HPCAL1, KRT17 and TRABD2A or Lam5, AQP4 and FREM3 (c.f., Maynard et al., 2021).

2) SiGra identifies RELN40 in L1– but the cited paper seems to refer to temporal lobe and not DLPFC. Experimental validation is mandatory to validate these claims.

3) The authors show that e.g., BayesSpace misidentifies the neutrophile as lymphocytes, meanwhile, it incorrectly identifies some tumor cells as myeloid cells or neutrophile. Seurat fails to disentangle epithelial cells from tumor cells... What type of errors is SiGra making when compared to ground truth?

4) Is the extend of lipofuscin and autofluorescence affecting the performance of SiGra?

5) Incidentally, I wonder whether SiGra could also help addressing inconsistencies in dissections (specifically considering gyrification of the human cortex). Could this even be a concern when using SiGra?

6) In figure 6 only 2 of the 12 slices are shown. Please provide criteria for selecting these slices out of the 12.

Some additional minor points:

- All abbreviations should be explained when used first in the manuscript
- Reference 2 and 42 are identical

RESPONSE TO REVIEWERS' COMMENTS

Reviewer #1 (Remarks to the Author):

This manuscript describes a new method, SiGra, for spatial domain detection and gene expression enhancement for single-cell spatial transcriptomics data. SiGra utilizes three graph transformer autoencoders to integrate the gene expression with image features, then performs spatial clustering and gene expression imputation using the joint embeddings. The paper is clearly written, and I appreciate the large amount of work devoted to this manuscript. However, I am concerned with the ad hoc nature of the method and its effectiveness in practice. In the spatial domain detection task, there is no theoretical justifications for almost all steps of the proposed model. In addition, the model in SiGra is very similar to a published method, MUSE, ([https://www.nature.com/articles/s41587-022-01251-z\[nature.com\]](https://www.nature.com/articles/s41587-022-01251-z[nature.com])), for the same purpose. In the gene expression enhancement task, SiGra does not produce any error control guarantee, either in the form of false discovery control or family-wise error rate control, making it hard to use in practice. No simulations were provided to help build up intuition and understand the performance of SiGra in different settings. My main concerns are:

1. The proposed method is very ad hoc in nature, with no justifications provided for many key steps of the algorithm. The main components of the SiGra are three autoencoders to extract image and gene expression embeddings. Why do we need separate imaging-based and transcriptomics-based autoencoders since the hybrid autoencoder already contains both gene expression and image information? The three autoencoders take data from different modalities as input, but why all of them are used to reconstruct gene expression? How does the attention mechanism balanced outputs from the three autoencoders? Is the final result dominated by embeddings from one autoencoder? The final loss function has three components and two hyper-parameters, λ_1 and λ_2 , are used to weight the gene loss and image loss. Why λ_1 and λ_2 are both set to 0.1? Does that mean the final loss is dominated by the hybrid loss only? Ablation study is needed to investigate the contribution of different components to the model.

Response: Thanks. We appreciate the reviewer's comment. In the revised manuscript, we added ablation studies to investigate the contributions of the transcriptomics-based encoder-decoder (G-ED), the image-based encoder-decoder (I-ED), and the hybrid encoder-decoder (H-ED) on both single-cell spatial transcriptomics (NanoString CosMx SMI) and spot-level spatial transcriptomics (10x Visium) data. We also provided the fine-tuning results for λ_1 and λ_2 , which were omitted in the original manuscript. We made revisions in our manuscript (page 9) and Supplementary Note 6. Details are shown below:

Ablation studies:

We presented the performance of three ablated models, with only one encoder-decoder in each model. As shown in the figure below: **1) NanoString CosMx SMI.** We first evaluated the adjusted rand index (ARI) scores achieved by the ablated models on the NanoString profiled lung cancer tissue slice. SiGra obtained the best ARI across all 20 field-of-views (FOVs) (median ARI: 0.59) than the other ablated models. In contrast, without the other two components, the hybrid encoder-decoder (H-ED) alone only achieved a median ARI of 0.34. The G-ED and I-ED also presented

lower ARI scores with median values of 0.40 and 0.24. **2) 10x Visium.** Then we evaluated the ARI scores achieved by the ablated models across the 12 dorsolateral prefrontal cortex (DLPFC) slices. SiGra obtained the highest ARI in all slices (median ARI: 0.57). In contrast, the H-ED presented a lower median ARI score of 0.33. The G-ED and the I-ED achieved median ARIs of 0.41 and 0.25, respectively.

To address the reviewer’s question of why an image-to-gene encoder-decoder (I-ED) instead of an image-to-image auto-encoder (I-AE) was used in SiGra, we further compared the performance of the I-AE-based and the I-ED-based ablated models. As shown in the figure below: **1) NanoString CosMx SMI.** Based on the lung cancer tissue slice, I-AE obtained much lower ARI (median: 0.04) than I-ED (median: 0.24) across different FOVs. **2) 10x Visium.** Based on the 12 DLPFC slices, I-AE also presented lower ARI (median: 0.14) than I-ED (median: 0.25). This ablation study suggested that the general imaging features extracted by an image-to-image autoencoder (I-AE) were less relevant to spatial domain detection. The imaging features that were relevant to the spatial gene expression patterns, which were extracted by the image-to-gene encoder-decoder (I-ED), contributed more to spatial domain identification. Additionally, the advantage of using an image-to-gene encoder-decoder rather than an image-to-image autoencoder was more significant for single-cell spatial transcriptomics data.

Collectively, these results demonstrated that the superior performance of SiGra was achieved through three encoder-decoder components. The ablated models with only the hybrid encoder-decoder H-ED or the other encoder-decoder alone were not sufficient to achieve comparable performance.

Fine-tuning for λ_1 and λ_2 :

Regarding the final loss function, we had two hyper-parameters, λ_1 and λ_2 , that were used to weigh the image-based loss and gene-based loss. In our manuscript, we identified the choice of λ_1 and λ_2 based on a grid-search approach. Through this grid-based hyper-parameter fine tuning, we identified that: for single-cell spatial transcriptomics data, the optimal parameters are $\lambda_1 = 0.1$ and $\lambda_2 = 0.1$; for 10x Visium data, the optimal parameters are $\lambda_1 = 1$ and $\lambda_2 = 1$. The fine-tuning results for these two weighting parameters were added to Supplementary Note 5 and as follows:

The two hyper-parameters, λ_1 and λ_2 , were used to balance the contributions of the three encoder-decoders through the image-based loss $L_{M,i}$ and gene-based loss $L_{g,i}$ relative to the hybrid loss $L_{h,i}$. Since the single-cell spatial transcriptomics data and the 10x Visium data were significantly different in terms of the image types (IHC vs H&E images, with different biological meanings of channels), spatial resolutions (single-cell level vs spot level), the coverage of the transcriptome (~1,000 genes vs the whole transcriptome), and the gene expression identification methods (probe-based spatial molecular imaging vs next generation sequencing), we fine-tuned the two hyper-parameters specifically for each data type.

We first performed coarse searches to identify the optimal parameter range for each data type, then used a grid-search approach for fine-tuning to determine the optimal values. The best options for the two parameters were chosen based on the loss evaluation on the validation set (30% of the overall data).

For single-cell transcriptomics data, the coarse search suggested that the optimal solution should be in the range between 0 and 1 for both parameters. Based on the 20 FOVs across the lung cancer tissue, we screened the options for λ_1 and λ_2 , ranging from 0.1 to 0.9 with a 0.2 interval. As shown in the figure below, the combination of $\lambda_1 = 0.1$ and $\lambda_2 = 0.1$ showed the lowest loss (median: 1.23) across all FOVs. Meanwhile, the combination of $\lambda_1 = 0.5$ and $\lambda_2 = 0.3$ showed the worst loss (median: 1.42).

For the 10x Visium data, the coarse searching suggested that the optimal solution should be in the range between 0 and 2 for both parameters. We chose the best options for the two parameters based on the loss evaluation on the validation set (30% of the overall data). Based on the 12 DLPFC slices, we screened the options for λ_1 and λ_2 , ranging from 0 to 2 with a 0.2 interval. As shown in the figure below, the combination of $\lambda_1 = 1$ and $\lambda_2 = 1$ showed lowest loss (median: 0.185) across all slices, while $\lambda_1 = 0$ and $\lambda_2 = 0$ showed the highest loss (median: 0.204) for all slices.

The fine-tuning results further suggested that all three encoder-decoders played important roles in reconstructing the spatial gene expression. For example, for the NanoString CosMx data, although the weights of the losses associated with the image-based encoder-decoder (I-ED) and gene-based encoder-decoder (G-ED) were both 0.1, these two encoders boosted the overall ARI from 0.34 (the hybrid encoder-decoder alone) to 0.57 (SiGra).

2. There are so many parameters in the model, how do the authors determine their optimal choices? The manuscript described the hyper-parameters from line 355 to 362 but did not explain how the optimal set was eventually chosen. Did the authors pick up one that gives the best results? If so, this clearly would lead to model over-fitting and overly optimistic results.

Response: Thanks for the reviewer's comment. In our manuscript, we selected the hyper-parameters including the embedding dimensions based on the grid-search approach. Similar to the selection of λ_1 and λ_2 , we fine-tuned the dimensions (D_1 and D_2) of the 1st and 2nd layers respectively, based on the loss obtained from the validation set (30% of the overall data). We screened the options for D_1 ranging from 128, 256, 512, and 1024, and D_2 ranging from 20, 30, 40, and 50.

Specifically, for the NanoString lung cancer slice, as shown in the figure below, the combination of $D_1 = 512$ and $D_2 = 30$ showed the lowest loss (loss: 1.240).

For the 10x Visium DLPFC slices, as shown in the figure below, the combination of $D_1 = 512$ and $D_2 = 30$ also showed the lowest loss (loss: 0.190).

3. The proposed model in this paper is very similar to a recently published method, MUSE (<https://www.nature.com/articles/s41587-022-01251-z> [nature.com]). MUSE also used multiple autoencoders to integrate the gene expression and image information for spatial domain detection, and they performed a detailed ablation study to explain the contribution of each component in their model. The authors should cite and compare with MUSE.

Response: Thanks for the comment. In the revised manuscript, we added comparisons with MUSE and cited it. We made related revisions in the manuscript (page 9) and Supplementary Note 6. Details are shown as follows:

As shown in the figure below, we evaluated the ARI scores achieved by different methods, including MUSE. Across all 12 DLPFC slices¹, SiGra obtained higher ARI (median ARI: 0.57) than the other methods, including stLearn (median ARI: 0.39), SpaGCN (median ARI: 0.40), and BayesSpace (median ARI: 0.44). In contrast, MUSE presented lower ARI scores with the median ARI of 0.31. The highest ARI that MUSE obtained was 0.43 on slice 151669, and the lowest ARI (ARI: 0.23) was on slice 151576. These results showed that MUSE had modest performance in recognizing the spatially organized brain structures.

Additionally, we included simulation studies to further compare the performance of MUSE with SiGra. SiGra also showed better performance than MUSE on simulation data. Full details were shown in our response to question #8, as well as our revised manuscript (page 9) and Supplementary Note 6.

4. The evaluation of spatial domain detection is inappropriate. Spatial domains are regions that show coherence in gene expression and biological function. A spatial domain may contain cells from multiple cell types. In Figure 2a, the authors performed spatial domain detection using SiGra and other methods, then compared the predicted spatial domains with annotated cell types using ARI. This is incorrect since the goal of SiGra and other comparing methods is domain detection, not cell-type identification.

Response: We appreciate the reviewer's comment. SiGra can identify spatial domains at various resolutions, depending on the data type and the applications. The spot-level spatial data has a low spatial resolution and consists of mixed cells/cell types in each spot. For example, the spatial resolution of the 10x Visium data is 100 μ m, measured between the centers of two neighboring spots. For such low-resolution data, SiGra directly and accurately reveals the spatial structures such as the anatomic layers in the DLPFC slices (Fig. 5) by clustering the latent-represented spots using Leiden. In contrast, the single-cell spatial data has significantly higher resolution. For example, the spatial resolution of the NanoString CosMx molecular imaging is 52nm, and the summarized gene expression profile based on image segmentation provides single-cell level resolution. SiGra thus can reveal spatial regions at the cellular level (Fig. 2 and 3) and microanatomic level (Fig. 4, the identifications of the microanatomic regions in the liver).

Meanwhile, on such high-resolution single-cell spatial data, the regional anatomic spatial domains can be revealed by further summarizing the Leiden clustering results (heterogenous cell types) with a dimensional moving window agglomeration approach. Such approaches have been well-

established in spatial data analysis of geographical information systems (GIS) data^{2,3} and have recently been used for revealing spatial domains in single-cell spatial data (for example, SSAM⁴ by Park et al.). Specifically, the SiGra clustering results were summarized by a circular window of diameter d sliding at both x and y directions across the whole image with a given stride length s . At each stop $C_{i,j}$ with the coordinate (x_i, y_j) , a vector $c_{i,j} \equiv [q_1, \dots, q_t]$ representing the proportions of the SiGra identified clusters (t) covered by the sliding window was calculated. All the stops $\{C_{i,j}\}$ were recursively merged to k groups $\{a_1, \dots, a_k\}$ by hierarchical clustering according to the cluster proportion vectors $\{c_{i,j}\}$. These agglomerated groups were defined as the discovered spatial domains. The original slide image was then labeled with the discovered spatial domains according to the coordinate of each stop. In this way, we obtained the spatial domains based on the heterogeneous cells identified on the spatial slice. The window radius d used in our work was $100\mu\text{m}$, which was consistent with the $10\times$ Visium spatial resolution, with the stride s of $10\mu\text{m}$.

The ground truth of the anatomic spatial domains of the NanoString CosMx lung cancer slide was provided by a certificated pathologist at Indiana University Health, Dr. Tieying Hou, according to the IHC images. As shown in the figure below, three spatial domains were identified by Dr. Hou: the tumor region (green), the desmoplasia region (red), and the adjacent normal region (orange). For fair comparisons with other methods, the same moving window agglomeration approach was used. Compared with the ground truth, SiGra achieved an ARI of 0.60, better than other methods including BayesSpace (ARI: 0.25), spaGCN (ARI: 0.10), Seurat (ARI: 0.10), stLearn (ARI: 0.10), and scanpy (ARI: 0.17). These results showed that SiGra obtained reliable spatial domains based on its identified accurate cell identities. It also indicated that the NanoString CosMx profiled cancer tissue slice was much more challenging given its strong cellular heterogeneity, large cell number, and high-resolution, compared with the $10\times$ Visium profiled normal DLPFC tissues which have well-organized anatomic structure.

To further verify the comparison results, we also tested BayesSpace and spaGCN for direct spatial domain identification of the three domains, without using the moving window agglomeration approach. BayesSpace and spaGCN only obtained ARIs of 0.15 and 0.19, respectively. These results further demonstrated that, for detecting large-scale anatomic spatial domains from single-cell spatial data, it was necessary to agglomerate the high-resolution cellular-level clustering results.

5. Another main application of SiGra is to enhance gene expression for imaging-based ST data. The authors stated that the rationale of this application is such type of data suffers from “missing values, data sparsity, low coverage, and noises” (line 50). However, I cannot agree with this statement. Although the processed gene expression data from CosMx and MERSCOPE are presented as a cell by gene matrix, the raw gene expression is measured using fluorescence imaging with subcellular resolution. The raw data is already very accurate and is in super high resolution. Without further evidence, I do not think there is any need for gene expression enhancement for such type of data.

Response: We appreciate the reviewer’s comment. Here we included the reasons for enhancing the single-cell spatial transcriptomics (SCST) data provided by current technologies. Though NanoString CosMx⁵ and MERSCOPE⁶ mentioned that their captured RNA transcripts had high specificity, the major limitations of such SCST data were investigated in the Jonathan L, et.al.⁶ study. Specifically, there were several factors resulting in the “missing values, data sparsity, low coverage, and noises”, including 1) cell segmentation and 2) RNA molecular overcrowding.

- 1) As shown in the Figure 3A and 3E of Jonathan L, et.al.⁶, the cell segmentation quality varied. Some cells had reliable segmented boundaries while others presented aberrant segmented boundaries, which were due to low-quality cell boundary and/or DAPI staining signals. Moreover, not all detected RNA transcripts lay inside segmented boundaries. Jonathan L, et.al.⁶ reported that the typical percentage of transcripts assignable to cells was only around 30-50%. Besides mRNAs that were truly located outside cells (such as those located inside exosomes), a significant number of cellular mRNAs could not be attributed to cells. Though a minimum cutoff of total RNA transcript count per cell of 20 was established to filter out cells with sparse statistics, cells remained in the SCST data also presented extensive sparseness.
- 2) Jonathan L, et.al.⁶ also reported that the gene panel used for SCST profiling contained some highly abundant genes in certain tissues, such as *C1qc* and *Gpx3* in mouse liver and kidney respectively. For cells containing large numbers of transcripts (i.e., over 1,000 per cell), the smFISH spots in the MERFISH images were too dense due to the overcrowding of RNA molecules, which prevented the accurate identification of single RNA transcripts. Moreover, the MERFISH transcript distributions were shown to be agreed with scRNA-seq data only for cells with low overall transcript counts, but not for cells with high overall transcript counts, which further indicated the noise and sparseness of SCST data. In addition, Jonathan L, et.al.⁶ examined the dropout rate between MERFISH and scRNA-seq data in their Figure 4, where the distribution of dropout rates was slightly shifted to higher dropout rates in MERFISH compared to scRNA-seq. It was also due to the molecular crowding effects that led to a high density of the fluorescent signal exceeding the diffraction limit, which could not be resolved and hence prevented those transcripts from being counted. Even for the detected signals, some were technically “dropped out” due to ambiguous signals. The NanoString CosMx protocol⁵ also removed detected transcripts according to whether all the four reporter probe-binding events happened within a radius of 90nm, and whether the mapping to the gene barcode is unique. The dropout rate at this step alone was 3 to 10%. Besides the transcript-level dropout, the cellular dropout rate was about 3% for cells containing less than 20 transcripts.

To further demonstrate that our enhanced data provided more information than raw data and to highlight the necessity of gene expression enhancement for such SCST data, inspired by these studies^{5,6}, we compared both enhanced data and raw data with existing bulk RNA-seq data. Specifically, 1) for the lung cancer NanoString CosMx data in our Fig. 2, we used the bulk RNA-seq data from TCGA lung cancer patients. 2) For the mouse liver MERSCOPE in our Fig. 3, we used the bulk RNA-seq data from The Tabula Muris Consortium⁷.

- 1) As shown in **Figure a** below, the x-axis and y-axis represented the total log-transformed counts per gene in lung cancer slices between the two types of technologies, i.e., SCST and bulk. Each point represented the RNA count for a single gene, averaged across different experimental samples for the corresponding technology. The RNA counts between enhanced SCST and bulk sequencing showed better concordance ($cor = 0.631$) than that between raw SCST and bulk data ($cor = 0.579$).
- 2) **Figure b** below showed the comparisons between SCST and bulk data for mouse liver MERSCOPE data, where the x-axis and y-axis were the total log-transformed counts. Similarly, each point represents the RNA count for a single gene, averaged across different samples for the corresponding technology. Again, we observed much higher correlation of RNA counts

between enhanced SCST and bulk data (cor = 0.854), in contrast with the comparisons between raw SCST and bulk data (cor = 0.800).

The above comparisons supported that our enhanced data provided more information than raw data and improved the data quality in SCST profiles. Related revisions were made in the manuscript (page 4-5) and Supplementary Note 1.

6. Figure 3 shows many ligand-receptor pairs with low correlations in the raw data have much stronger correlations in SiGra enhanced data. The authors wanted to use this as evidence to support the usefulness of the enhanced data by SiGra. In my experience, many L-R pairs are loosely correlated due to their relatively low expression level. I doubt whether the strong correlations observed in the enhanced data are true biological interactions or just false discoveries. The authors did not provide any solid evidence to support their findings.

Response: Thanks for the reviewer's comment. In the revised manuscript, we evaluated the potential false discoveries in the L-R associations using randomized control. Briefly, we assumed that randomly selected gene pairs from the SCST data were not likely associated and thus used as negative controls. By comparing with these negative controls, the false discovery rate of the selected L-R associations was estimated. Briefly, 10,000 gene pairs were randomly selected, and the corresponding Pearson correlations were calculated as negative control. For each of the L-R pair examined, we estimated the false discovery rate and updated Fig. 3 accordingly using the FDR values based on the negative controls instead of the Pearson correlations.

For the NanoString CosMx non-small cell lung cancer dataset (Fig. 3), as shown in the **Figure a** below, the y-axis and x-axis referred to the FDR value of each L-R pair in the enhanced and raw data respectively. Across the total 660 L-R interactions, 55 L-R pairs from the enhanced data were statistically significant ($FDR < 0.05$), whereas 42 L-R pairs from the raw data had $FDR < 0.05$. There were 28 L-R pairs shared between enhanced data and raw data, indicating enhanced data preserved useful information of raw data. In addition, 27 specific L-R interactions were identified from the enhanced data, while 14 specific L-R interactions were found in the raw data. We further investigated whether these specific L-R interactions pairs were similar with the shared L-R pairs. For those significant L-R pairs identified from the enhanced data, there were no significant differences between the specific and shared L-R pairs (**Figure b**), suggesting that both had similar probability of being true associated L-R pairs. In contrast, the raw-specific L-R pairs were significantly different from the shared L-R pairs (**Figure c**), suggesting that the raw-specific pairs were more likely to be false discoveries than the shared L-R pairs. These results indicated that the enhanced data not only enabled to detect more L-R interactions than the raw data, but also the identified L-R pairs were more likely to be true discoveries than those specifically detected in raw data. The data enhancement using SiGra not only improved the sensitivity of L-R interaction detection (identifying more L-R pairs), but also preserved the specificity (the specifically identified L-R pairs that had similar statistical significance with the shared L-R pairs). Related revisions were made in the manuscript (page 5) and Supplementary Note 1.

7. Following my above concern, the analysis of the differentially expressed gene using the enhanced data suffers from the same problem.

Response: Thanks for the reviewer’s comment. Based on our above explanation and results, we anticipated that the differential expression analysis could benefit from the enhanced data given its improved data quality. To further verify it, regarding our Fig. 4e, we used the single-cell RNA-seq data from the Tabula Muris Consortium 2020⁸ to identify the DEGs in the cell clusters of hepatocytes, periportal hepatocytes, hepatic stellate cells, and endothelial cells. In this way, we then evaluated the overlaps between the scRNA-seq’s DEGs and enhanced data’s DEGs, as well as the overlaps between the scRNA-seq’s DEGs and the raw data’s DEGs.

As shown in the figure below, we identified the overlapped DEGs with scRNA-seq for each cluster. The purple-colored bars represent the number of DEGs shared between scRNA-seq and raw data, and the orange-colored bars represented the number of DEGs shared between scRNA-seq and enhanced data. We also labeled the number of DEGs on the bar plot, for example, for C-1, “12 vs 12” referred to “the shared DEGs between scRNA-seq and raw data” vs “the DEGs of raw data”, and “49 vs 59” referred to “the shared DEGs between scRNA-seq and enhanced data” vs “the DEGs of enhanced data”. Across different clusters, the enhanced data was shown to recover more dysregulated genes than the raw SCST data.

8. No simulations were performed in the study. Simulations would be very helpful in understanding some basic properties of SiGra and its performance in different settings as compared to the other methods.

Response: Thanks for the reviewer’s comment. In this revision, we performed the simulation comparisons based on the simulation design of MUSE¹⁶. The only difference between our simulated data and MUSE’s design was that, the simulation data were generated with spatial locations for each domain.

Details of simulation steps were as follows:

1) First, we generated the ground truth of domain regions $l \in \{1, 2, \dots, L\}$, and K different cell types, where L was the number of spatial domains and K was the number of cell types ($K \geq L$). Each domain was a spatial rectangle region $R_l = \{(r_x, r_y); s_{x0} < r_x < s_{x1}, s_{y0} < r_y < s_{y1}\}$. In each domain, we assigned a dominating/major cell type, with several other cell types scattered and mixed with the major one in this spatial domain. For each cell, we randomly generated its spatial coordinate (s_x, s_y) , where $s_{x0} < s_x < s_{x1}, s_{y0} < s_y < s_{y1}$. Here we simulated four spatial regions: $R_1 = \{c-1, c-5, c-6\}$, $R_2 = \{c-2\}$, $R_3 = \{c-3, c-7, c-8\}$, $R_4 = \{c-4, c-9\}$. The dominating cell types c-1 in R_1 , c-2 in R_2 , c-3 in R_3 , and c-4 in R_4 had more cell numbers than the other cell types $\{c-5, c-6, c-7, c-8, c-9\}$. For each dominating cell type, we generated 1,000 cells in their respective spatial region. For the other mixed cell types, we generated 300 cells respectively.

2) Next, the latent representations of gene expression and morphology images features: Z_G, Z_I ($Z_G \in R^m, Z_I \in R^m$), were generated following the design of MUSE, where m was the size of the latent dimension. That is, for either Z_G or Z_I , the latent representations Z_i of the i -th cell was simulated using a multivariable normal distribution: $Z_i \sim \sum_l^L \pi_{l,i} MVN(\mu_l, \Sigma_l)$, where L was the total number of the domain regions ($L = 4$). If cell i belonged to the l -th domain, then $\pi_{l,i} = 1$, otherwise $\pi_{l,i} = 0$. $\mu_l \in R^m$ was sampled from a uniform distribution with $\Sigma_l \in R^{m \times m}$ the identity matrix. Note that the latent gene features Z_G and latent image features Z_I were generated separately, that is:

$$Z_{iG} \sim \sum_l^L \pi_{l,i} MVN(\mu_{lG}, \Sigma_{lG}); Z_{iI} \sim \sum_l^L \pi_{l,i} MVN(\mu_{lI}, \Sigma_{lI})$$

3) The raw gene expression and image features were then generated by a linear transformation by $X_G = A_G Z_G + \delta$ and $X_I = A_I Z_I + \delta$, where $A \in R^{p \times m}$ was the random projection matrix from the uniform distribution between $[-0.5, 0.5]$. δ was the gaussian noise sampled from $N(0, \sigma^2)$. With additional dropouts added as below, the raw gene expression X'_G and image features X'_I were obtained:

$$X'_G = X_G \mathbf{1}[\exp(-\alpha_G X_G) < \eta_G]; X'_I = X_I \mathbf{1}[\exp(-\alpha_I X_I) < \eta_I]$$

, where $\mathbf{1}[\cdot]$ was the indicator function, which returned 1 if the argument was true, otherwise returned 0. α was the decay coefficient that controlled dropout levels and η was the random value sampled from the uniform distribution between $[0, 1]$.

With the simulated data X'_G and X'_I , we used them as input for simulation experiments and benchmarking.

Herein, we generated simulation data including both gene-based and image-based features. We chose three settings with different dropout levels for simulation data, i.e., α as 0.2, 0.3, and 0.4.

- When $\alpha = 0.2$, the generated data was visualized as below in **Figure a**. As described above, the spatial data consisted of four spatial regions, where each of them was dominated by one major cell type with other scattered types of cells. Meanwhile, the gene-based features and image-based features were shown to contribute to spatial domain identification at certain levels, with ARI as 0.54 and 0.40 respectively. However, MUSE failed to reveal

clear spatial domains with ARI only as 0.28. In contrast, SiGra achieved higher ARI (ARI: 0.71) revealing much more accurate spatial domains.

- When $\alpha = 0.3$, the generated data was visualized in **Figure b**. Consistently, four spatial domains were generated, with both gene-based features and image-based features contributed to spatial domain identification (ARI: 0.69, 0.60) respectively. Nevertheless, MUSE only obtained ARI as 0.42, which was much lower than SiGra (ARI: 0.76).
- When $\alpha = 0.4$, the generated data was visualized in **Figure c**. Similarly, both gene-based features and image-based features contributed to spatial domain identification at certain levels (ARI: 0.75, 0.67). SiGra maintained more accurate (ARI: 0.77) than MUSE (ARI: 0.55) in revealing spatial domains.

In addition, for each setting, i.e., the dropout levels α was 0.2, 0.3, and 0.4, we generated 10 replicated simulation data and obtained the boxplot of ARI scores for gene-only, image-only, MUSE, and SiGra, respectively. As shown in the figure below, SiGra reached much higher ARI scores (median: 0.723) than MUSE (median: 0.24) when the dropout levels $\alpha = 0.2$. For α was 0.3 and 0.4, SiGra also presented higher ARI (median: 0.77; 0.78) than MUSE (median: 0.39; 0.51). These simulation results showed that SiGra achieved more accurate identification of spatial regions through leveraging both gene-based and image-based features. Related revisions were made in the manuscript (page 9) and Supplementary Note 6.

Reviewer #2 (Remarks to the Author):

(Please see attached document.)

The authors developed Graph Neural Networks based algorithm for spatial domains recovery and transcriptomics data denoising. The authors applied the proposed algorithm to several real datasets and showed some promising results. This is an very interesting work, but there several important issues that need to be addressed. These can be summarized as follows:

Major concern:

- The terminology of graph transformer convolutional layer is misleading. Multi-head attention within the neighborhood has been widely used in the graph neural networks research area. This kind of approach is called "Graph Attention Networks" or "GAT", instead of "Graph Transformer" or " Graph Transformer Convolutional Layer". The crucial difference between these two lies in the way to model graph structural information. "GAT" is a very common and popular graph neural network model. It propagates the node features to the first-order neighbors via attention and gradually extends to high-order neighbors by stacking the graph attentional layers, which is similar to other GNNs, e.g. GCN, GraphSAGE. In contrast, "Graph Transformer" is another family of models that densely connects all the nodes in the graph and use positional encoding or graph neural networks to encode structural information.

Response: Thanks for the reviewer's comment. The definition of "Graph Transformer" is fast evolving, reflecting the exciting and active advances in this field. Here we reviewed recently published graph transformers that were similar to SiGra in the three perspectives: using a fully connected subgraph instead of the whole graph, omitting positional encoding, or using a multiple layer convolution strategy. We summarized the hallmark characteristics of a transformer model (self-attention and fully connected nodes in a subgraph or a graph) and showed that SiGra has all such characteristics. We also showed that other features are optional. We concluded that SiGra was qualified as graph transformer-based model. We further explained that the architecture of SiGra was common in graph transformer field.

In the original transformer paper⁹, Vaswani et al defined the two hallmark characteristics of a transformer as self-attention and the full connections of all positions in both encoders and decoders. When adapting the original transformer in large graph-structured data with sub-million-level to billion-level nodes, it was computationally forbidden to include the whole graph and build full-connection layers among all nodes. In such scenarios, node-specific subgraphs and convolution across all nodes were often used in graph transformer models. Among options of subgraphs, the nearest neighbor network was the most common choice, with other sampling approaches such as top-k intimacy sampling (Graph-BERT¹⁰). For example, in the graph transformer networks developed by Yun et al¹¹, at each Graph Transformer Layer, only attentions from the nearest neighbors for the corresponding edge type was considered, and a convolution was performed across all nodes to learn the shared weights. In the point transformer model¹² developed by Zhao et al for point-cloud data, the k-nearest neighbor network was used as the subgraph for attention learning. Dwivedi et al¹³ in the work for generalizing transformer networks to graphs, regarding large graphs, only local neighbors were used as subgraphs for attention learning. In the Unified Message Passing Model (UniMP), Shi et al¹⁴ used a graph transformer layer with only nearest neighbors as the subgraph for attention learning. Besides the adaptation of Vaswani's

vanilla transformer, the UniMP model also used a convolution architecture similar to GCN and GAT, with the convolution across all nodes in a graph transformer layer, and with multiple transformer layers to gradually propagate node labels from distant nodes to the node of interest.

SiGra can be considered as a typical graph transformer, as it uses similar architecture with the above graph transformer models. It qualifies the two hallmark characteristics: self-attention and the full connection of all nodes in each subgraph. The use of the nearest neighbor-based subgraphs in SiGra is also common in such transformer models as we discussed above. Moreover, positional encoding is optional but not necessary in transformers. Positional encoding is not a hallmark that distinguish transformers from GNNs. For example, Dwivedi et al¹⁵ used positional encoding in graph neural network models. As a graph transformer model, the other practical difference between SiGra and GAT is that SiGra uses the typical query-key-value structure for attentions, which is similar to most transformer models and different to GAT's attention mechanism.

- For the image layer, the name 'autoencoder' is also misleading. The input here is pixel vector and the output is gene expression not the image itself.

Response: Thanks for the reviewer's comment. We revised the name "autoencoder" to specific encoder-decoders, i.e., the imaging-based encoder-decoder (I-ED), the hybrid encoder-decoder (H-ED), and the transcriptomics-based encoder-decoder (G-ED). Related revisions were made in the manuscript (page 3, page 8).

- Still for the image layer, the authors vectorize the images and concatenate them as input. As we know, learning shapes or other topological characters from vectorizing the images is suboptimal. Can the authors provide some insights to support their choice? Also, for each cell, fixed sized square images are extracted as input. Since different cells vary in their sizes, which means many images contain a significant portion of background even info from neighboring cells. I am curious why those extra (noisy) part of image information did not significant affect the performance of the model? More importantly, I am curious the contribution of image to the model performance some ablation studies to illustrate the contribution of image and spatial graph.

Response: Thanks for the reviewer's comment.

The choice of vectorizing the multichannel images is a trade-off for generalizability, scalability, and computing time. SiGra trains the Image-Gene encoder-decoder for each new dataset instead of using a pre-trained image feature extractor. This makes SiGra generalizable to data generated from new tissue types and imaging data generated from samples staining by different protocols. Furthermore, single-cell spatial omics data fast evolves with large cell size, thus we prefer to use an Image-Gene encoder-decoder that can be trained efficiently. Besides our current choice, another option is to use a Visual Geometry Group (VGG) model. However, training a new VGG model for new datasets is computationally challenging and not practical for general users. Using pre-trained VGG models is also suboptimal, since such VGG models are pre-trained for natural object classification and are not optimized for extracting cell morphology features from IHC images. Therefore, we have decided to use a simpler image feature extractor via vectorization of each image and concatenation of images from multiple channels so that the Image-Gene encoder-decoder can be trained tailored and efficiently for new datasets.

The size of the square images is intentionally selected to cover the boundary regions of neighboring cells. The rationale is that the morphologies of both the cell of interest and its neighbors at the cell-cell interaction areas provide important biological information of both cell types and cell functions, and thus should be extracted into the SiGra model. For example, for a tumor-associated fibroblast infiltrating into the tumor region, the morphology of a neighboring tumor cell at the boundary region provides biological clues of the fibroblast-tumor interaction. Another option, as what MUSE does, is to use a cell mask to only include image contents within the identified cell body. We did not use this solution as it missed the important morphology information described above. Another concern is, using a cell mask will make the model very sensitive to cell segmentation results. This concern is even more noteworthy for cells with irregular boundary outlines, such as elongated fibroblasts, and in this case, cell segmentation is not accurate and challenging. Therefore, SiGra directly uses fixed sized square images that are not affected by the accuracy of cell segmentations. In summary, including the boundary regions of neighboring cells captures crucial biological information, as well as the direct use of the image information without relying on cell segmentations, allows SiGra obtaining robust and superior performance.

In the revised manuscript, we added ablation study to investigate the contributions of transcriptomics-based encoder-decoder (G-ED), imaging-based encoder-decoder (I-ED), as well as the hybrid encoder-decoder (H-ED), respectively. Here we presented the performance of three ablated models, with only one encoder-decoder in each model. As shown in the figure below: **1) NanoString CosMx SMI.** We first evaluated the adjusted rand index (ARI) scores achieved by the ablated models on the NanoString profiled lung cancer tissue slice. SiGra obtained the best ARI's across all 20 field-of-views (FOVs) (median ARI: 0.59) than the other ablated models. The hybrid encoder-decoder (H-ED) and the I-ED presented lower ARI scores with median values as 0.34 and 0.24. Specifically, for FOV 9, I-ED reached comparable ARI (ARI: 0.41) with G-ED (ARI: 0.40). **2) 10x Visium.** Then we evaluated the ARI scores achieved by the ablated models across the 12 dorsolateral prefrontal cortex (DLPFC) slices. SiGra obtained the highest ARI in all slices (median ARI: 0.57). The H-ED and I-ED presented lower median ARI scores of 0.33 and 0.25. Specifically, in slice 151675, I-ED obtained comparable ARI (ARI: 0.34) with G-ED (ARI: 0.38). In the other slices including 151509 and 151510, I-AE and G-AE also obtained similar ARIs. These results show that the image-based encoder-decoder (I-ED) indeed contributes to the SiGra model. Related revisions were made in the manuscript (page 9) and Supplementary Note 6.

- In Fig2, the authors used the cell label produced by He et al. as ground truth. However, according to the paper, Cell type was determined by comparing individual cells' expression profiles to reference profiles for different cell types (scRNA-seq and bulk RNA-seq of sorted blood and stroma databases), assigning each cell to the cell type under which its profile was most likely. The likelihood was defined using a negative binomial distribution, with mean defined by a cell type's reference profile plus expected background, and with a size parameter set at 10 to allow for extensive overdispersion. Namely, they used a likelihood based method to generate cell labels. It is not ideal to treat those labels as ground truth. Also, there are 8 slides in the CosMX website, can the author also include the result for the other 7 slides.

Response: Thanks for the reviewer's comment.

- 1) We did agree that the likelihood-based inference of cell types was not ideal as ground truth. Though some cells might be mistakenly labeled, it still provided reasonable and mostly likely cell identities on the spatial slide. For instance, tumor cells would not possibly be mistakenly identified as immune cells, or vice versa. As shown in our Fig. 2, SiGra provided very clear and accurate separation of tumor and immune cells, whereas other methods even failed to clearly distinguish such cell types.
- 2) Moreover, considering the likelihood-based inference of cell types might not be accurate for the identity of each individual cell, here we used the domain-based strategy to evaluate each method's performance. That is, given the heterogeneous cells existing in the lung cancer spatial slice, we tried to interrogate the spatial domains that consisted of mixed cell types. Such spatial domains would provide a macro perspective focusing on the major cell population rather than delving into individual cells, which made it minimally affected by the accuracy of the identities of a few individual cells.

To reveal the spatial domains in the single-cell spatial data, we used a sliding circular window approach to summarize the heterogeneous cells. This approach has been well-established in spatial data analysis of geographical information systems (GIS) data^{2,3} and has recently been used for revealing spatial domains in single-cell spatial data (for example, SSAM⁴ by Park et al.). Specifically, the SiGra clustering results (cell types) were summarized by a circular window of diameter d sliding at both x and y directions across the whole image with a given stride length s . At each stop $C_{i,j}$ with the coordinate (x_i, y_j) , a vector $c_{i,j} \equiv [q_1, \dots, q_t]$ representing the proportions of the SiGra identified clusters (t) covered by the sliding window was calculated. All the stops $\{C_{i,j}\}$ were recursively merged to k groups $\{a_1, \dots, a_k\}$ by hierarchical clustering according to the cluster proportion vectors $\{c_{i,j}\}$. These agglomerated groups were defined as the discovered spatial domains. The original slide image was then labeled with the discovered spatial domains according to the coordinate of each stop. In this way, we obtained the spatial domains based on the heterogeneous cells identified on the spatial slice. The window radius d used in our work was 100 μm , which was consistent with the 10x Visium spatial resolution, with the stride s as 10 μm .

As shown in the figure below, the ground truth of the anatomic spatial domains of the NanoString CosMx lung cancer slide was provided by a certified pathologist at Indiana University Health, Dr. Tieying Hou, according to the IHC images. Specifically, three spatial

domains were identified by Dr. Hou: the tumor region (green), the desmoplasia region (red), and the adjacent normal region (orange). For fair comparisons with other methods, the same sliding window agglomeration approach was used. Compared with this ground truth, SiGra achieved an ARI of 0.60, better than other methods including BayesSpace (ARI: 0.25), spaGCN (ARI: 0.10), Seurat (ARI: 0.10), stLearn (ARI: 0.10), and scanpy (ARI: 0.17). These results showed that SiGra outperformed current existing methods in obtaining reliable spatial regions.

3) Here we also provided the results for the other 7 slides.

We included all 8 lung cancer slices and evaluated the ARI scores achieved by different methods. As shown in the figure below, across all slides, SiGra obtained higher ARI (median ARI: 0.51) than the other methods, including stLearn (median ARI: 0.42), SpaGCN (median

ARI: 0.25), and BayesSpace (median ARI: 0.27). In the slice of lung-6 patient, SiGra achieved much better ARI (ARI = 0.7) than competitors. These results showed that SiGra outperformed the other methods in accurately recognizing cells in single-cell spatial data. In addition, we also visualized the spatial figures of SiGra based on the three slices of lung-5 patients. Related revisions were made in the manuscript (page 9) and Supplementary Note 6.

- I also notice that the authors did not include STAGATE for performance comparison (Deciphering spatial domains from spatially resolved transcriptomics with adaptive graph attention auto-encoder.) Can the authors also include STAGATE?

Response: Thanks for the reviewer’s comment. We included STAGATE and compared with it comprehensively, including using 1) simulation data, 2) 10x Visium data, and 3) the 8 NanoString CosMx lung cancer slices. Related revisions were made in the manuscript (page 9) and Supplementary Note 6.

1) Simulation data

Here we performed the simulation comparisons based on the simulation design of MUSE¹⁶. The simulation data were generated with spatial locations for each domain.

Details of simulation steps were as follows:

- First, we generated the ground truth of domain regions $l \in \{1, 2, \dots, L\}$, and K different cell types, where L was the number of domain layers and K was the number of cell types ($K \geq L$). Each domain was a spatial rectangle region $R_l = \{(r_x, r_y); s_{x0} < r_x < s_{x1}, s_{y0} < r_y < s_{y1}\}$. In each domain, we assigned a dominating/major cell type and several other cell types scattered and mixed with the major one in this spatial region. For each cell, we randomly generated its spatial coordinate (s_x, s_y) , where $s_{x0} < s_x < s_{x1}, s_{y0} < s_y < s_{y1}$. Here we simulated four spatial regions: $R_1 = \{c-1, c-5, c-6\}$, $R_2 = \{c-2\}$, $R_3 = \{c-3, c-7, c-8\}$, $R_4 = \{c-4, c-9\}$. The dominating cell types c-1 in R_1 , c-2 in R_2 , c-3 in R_3 , and c-4 in R_4 had more cell numbers than the other cell types $\{c-5, c-6, c-7, c-8, c-9\}$. For each dominating cell type, we generated 1,000 cells in their respective spatial region. For the other mixed cell types, we generated 300 cells respectively.
- Next, the latent representations of gene expression and morphology images features: Z_G, Z_I ($Z_G \in R^m, Z_I \in R^m$), were generated following the design of MUSE, where m was the size of the latent dimension. That is, for either Z_G or Z_I , the latent representations Z_i , of the i -th cell was simulated using a multivariable normal distribution: $Z_i \sim \sum_l^L \pi_{l,i} MVN(\mu_l, \Sigma_l)$, where L was the total number of the domain regions ($L = 4$). If cell i belonged to the l -th domain, then $\pi_{l,i} = 1$, otherwise $\pi_{l,i} = 0$. $\mu_l \in R^m$ was sampled from a uniform distribution with $\Sigma_l \in R^{m \times m}$ the identity matrix. Note that the latent gene features Z_G and latent image features Z_I were generated separately, that is:

$$Z_{iG} \sim \sum_l^L \pi_{l,i} MVN(\mu_{lG}, \Sigma_{lG}); Z_{iI} \sim \sum_l^L \pi_{l,i} MVN(\mu_{lI}, \Sigma_{lI})$$

- The raw gene expression and image features were then generated by a linear transformation by $X_G = A_G Z_G + \delta$ and $X_I = A_I Z_I + \delta$, where $A \in R^{p \times m}$ was the random projection matrix from the uniform distribution between $[-0.5, 0.5]$. δ was the gaussian noise sampled from $N(0, \sigma^2)$. With additional dropouts added as below, the input/raw gene expression X'_G and image features X'_I were obtained:

$$X'_G = X_G \mathbf{1}[\exp(-\alpha_G X_G) < \eta_G]; X'_I = X_I \mathbf{1}[\exp(-\alpha_I X_I) < \eta_I]$$

, where $\mathbf{1}[\cdot]$ was the indicator function, which returned 1 if the argument was true, otherwise returns 0. α was the decay coefficient that controlled dropout levels and η was the random value sampled from the uniform distribution between [0,1].

With the simulated data X'_G and X'_I , we used them as input for simulation experiments and benchmarking. For fair comparisons, we follow the default settings in STAGATE. Herein, we generated simulation data including both gene-based and image-based features. We chose three settings with different dropout levels for data generation, i.e., α as 0.2, 0.3, and 0.4.

As shown in the figure above, when $\alpha = 0.2$, the generated data was visualized in **Figure a**. As described above, the spatial data consists of four spatial regions, where each of them was dominated by one major cell type and mixed with other cell types. Meanwhile, the gene-based features and image-based features were shown to contribute to spatial domain identification at certain levels, with ARI as 0.54 and 0.40 respectively. SiGra achieved higher ARI (ARI: 0.71) with much more accurate spatial domains than STAGATE (ARI: 0.58). For $\alpha = 0.3$ (**Figure b**), similarly four spatial domains were generated, with both gene-based features and image-based features contributed to spatial domain identification (ARI: 0.69, 0.60) respectively. Nevertheless, STAGATE only obtained ARI as 0.61, which was lower than SiGra (ARI: 0.76). When $\alpha = 0.4$ (**Figure c**), SiGra maintained more accurate (ARI: 0.77) than STAGATE (ARI: 0.63) in revealing spatial domains.

In addition, for each setting, i.e., the dropout levels α is 0.2, 0.3, and 0.4, we generated 10 replicated simulation data and obtained the boxplot of ARI scores for gene-only, image-only, STAGATE, and SiGra, respectively. As shown in the figure below, SiGra reached much higher ARI scores (median: 0.723) than STAGATE (median: 0.573) when the dropout levels $\alpha = 0.2$. When α was 0.3 and 0.4, SiGra also presented higher ARI (median: 0.77; 0.78) than STAGATE (median: 0.60; 0.62). These simulation results showed that SiGra achieved more accurate identification of spatial regions through leveraging both gene-based information and image-based information.

2) 10x Visium data

As shown in the figure below, we evaluated the ARI scores achieved by different methods, specifically including STAGATE. Across all 12 DLPFC slices¹, SiGra obtained higher ARI (median ARI: 0.57) than the other methods, including stLearn (median ARI: 0.39), SpaGCN (median ARI: 0.40), and BayesSpace (median ARI: 0.44). In contrast, STAGATE presented slightly lower ARI scores with median ARI as 0.49. On slice 151669, STAGATE obtained lowest ARI (ARI: 0.27), where SiGra achieved much better ARI (ARI: 0.49). These results showed that SiGra outperformed STAGATE in recognizing the organized brain structures.

3) NanoString CosMx lung cancer slices

We included all 8 lung cancer slices and evaluated the ARI scores achieved by different methods, specifically including STAGATE. As shown in the figure below, across all slices, SiGra obtained higher ARI (median ARI: 0.51) than the other methods, including stLearn (median ARI: 0.42), SpaGCN (median ARI: 0.25), and BayesSpace (median ARI: 0.27). In contrast, STAGATE presented much lower ARI scores with median ARI only as 0.22. For the slice of lung-6, STAGATE only obtained ARI around 0.1, whereas SiGra achieved much better ARI (ARI: 0.7). These results showed that SiGra outperformed STAGATE in recognizing single-cell spatial data.

- For the L-R results in both Figures 3 and 4, can the authors also include some results to evaluate the potential false positive correlations generated by the algorithm using randomly selected two genes? Since SiGra integrates information from neighboring cells, I suspect it will also create some false correlation results.

Response: Thanks for the reviewer's comment. In the revised manuscript, we evaluated the potential false discoveries in the L-R associations using randomized control as suggested by the reviewer. Briefly, we assumed that randomly selected gene pairs from the SCST data were not likely associated and thus used as negative controls. By comparing with these negative controls, the false discovery rate of the selected L-R associations was estimated. Briefly, 10,000 gene pairs were randomly selected, and the corresponding Pearson correlations were calculated as negative control. For each of the L-R pair examined, we estimated the false discovery rate and updated Fig. 3 and Fig. 4 accordingly using the FDR values based on the negative controls instead of the Pearson correlations.

For the NanoString CosMx non-small cell lung cancer dataset (Fig. 3), as shown in the **Figure a** below, the y-axis and x-axis referred to the FDR values of each L-R pair in the enhanced and raw data respectively. Across the total 660 L-R interactions, 55 L-R pairs from the enhanced data were statistically significant ($FDR < 0.05$), whereas 42 L-R pairs from the raw data had $FDR < 0.05$. There were 28 L-R pairs shared between enhanced data and raw data, indicating enhanced data preserved useful information of raw data. In addition, 27 specific L-R interactions were identified from the enhanced data, while 14 specific L-R interactions were found in the raw data. We further investigated whether these specific L-R interactions pairs were similar with the shared L-R pairs. For those significant L-R pairs identified from the enhanced data, there were no significant difference between the specific and shared L-R pairs (**Figure b**), suggesting that both had similar probability of being true associated L-R pairs. In contrast, the raw-specific L-R pairs were statistically different from the shared L-R pairs (**Figure c**), suggesting that the raw-specific pairs were more likely to be false discoveries than the shared L-R pairs. These results indicated that the enhanced data not only enabled to detect more L-R interactions than the raw data, but also the identified L-R pairs were more likely to be true discoveries than those specifically detected in raw data. The data enhancement using SiGra not only improved the sensitivity of L-R interaction detection (identifying more L-R pairs), but also preserved the specificity (the specifically identified L-R pairs that had similar statistical significance as the shared L-R pairs).

For the Vizgen MERSCOPE mouse liver dataset (Fig. 4), as shown in the figure below, the y-axis and x-axis referred to the FDR values of each L-R pair in the enhanced and raw data respectively. Among the 64 L-R pairs identified in this dataset, 13 L-R pairs from the enhanced data were statistically significant ($\text{FDR} < 0.05$), whereas 12 L-R pairs from the raw data had $\text{FDR} < 0.05$. There were 9 L-R pairs shared between the enhanced and the raw data, indicating the enhanced data preserved useful information of the raw data. In addition, enhanced data had 4 specific L-R interactions, while raw data had 3 specific L-R interactions. We further examined the L-R interactions specifically identified from the enhanced and the raw data, respectively, using the bulk RNA-seq data from the Tabula Muris Consortium⁷ as the validation dataset. The 4 L-R pairs specifically identified from the enhanced data also presented strong correlations in the validation dataset (Wnt2-Fzd4: 0.581; Pkm-Cd44: 0.885; Col1a2-Itga2b: 0.641; Dll1-Notch2: 0.798). However, the raw-specific L-R pairs showed low correlations in bulk data (Fgf1-Egfr: 0.386; Timp3-Kdr: 0.498; Jag1-Notch1: 0.115). These results indicated that those raw-specific L-R pairs were more likely to be false discoveries, which could result from noises and the low data quality in the raw data. Therefore, SiGra improved both the sensitivity (more identified L-R pairs) and the specificity (more true discoveries) for the detection of L-R interactions.

Minor concerns:

- For Fig 3 A, why the tumor cells are not included?

Response: Thanks for the reviewer's comment. For visualization clarity, tumor cells were not shown in **Fig. 3a**. We have the tumor cells included in the UMAP at the **Supplementary Fig. 1b**.

- In Fig 4, the authors claim 'For example, Cd34 and Vwf show scattered false signal...'. Do the authors have any evidence to support this claim that the expression of Cd34 and Vwf on those cells are false?

Response: Thanks for the reviewer's comment. In the liver tissue, Vwf is restrictively expressed in endothelia cells and megakaryocytes, but not in other cell types^{17,18}. Mature megakaryocytes are located near blood vessels in the liver tissue¹⁹. Therefore, the expression pattern of Vwf in the liver tissue (**Supplementary Fig. 2d**) should be aligned with the blood vessels. Cd34 expression in healthy liver tissue mainly expressed in the liver sinusoidal endothelial cells²⁰ and also aligned well with blood vessels in healthy livers. Cd34 is more likely to show in the other regions during the pathologic capillarization in cirrhosis and other cases. Therefore, the Vwf expression outside the vessel regions in the liver tissue is likely to be the false signals. The expression of Cd34 away from the vessel regions in healthy liver tissue is also questionable.

Reviewer #3 (Remarks to the Author):

I had the chance to review an interesting manuscript that describes a new technique for the improved assessment of spatial transcriptomic data. Single-cell spatial elucidation through image-augmented GRaph transformer (SiGra) incorporates multi-modal spatial features in a more effective manner – using graph transformer autoencoders with attention mechanism – and improves the limits that raw data faces to decipher spatial domains. SiGra is reportedly one of the first methods to utilize multi-channel images in a highly effective manner (true?).

SiGra is benchmarked and compared to other methods of analysis, namely Seurat v4, Scanpy, stLearn, SpaGCN, and BayesSpaces based on previously published datasets published using NanoString CosMx SMI, MERSCOPE, and 10X Visium.

While the study appears for the most part rigorously executed and analyzed - personally, I am convinced that it is worth trying SiGra – at this point most of the results represent rather indirect evidence that SiGra is indeed superior in unraveling new and important biological mechanisms.

More direct evidence including some experimental validation for their claims is required to justify publication.

1) The results in the DLPFC in this study must be carefully compared to Maynard et al., 2021 – who used gene-level statistics from both layer-specific versus cell type-specific expression profiles for “spatial registration” in their study. What about HPCAL1, KRT17 and TRABD2A or Lam5, AQP4 and FREM3 (c.f., Maynard et al., 2021).

Response: Thanks for the reviewer’s comment. Based on the results of DLPFC in our study, here we carefully compared the layer-enriched gene markers (HPCAL1, KRT17, TRABD2A, LAMP5, AQP4, FREM3) in our enhanced data with the original study¹ (Maynard et al., 2021). Specifically, we performed the exact statistical analysis in Maynard et al 2021 (“Layer-level gene modeling” and fit ‘Enrichment’ and ‘Pairwise’ models) using the enhanced data obtained by SiGra. The variations of gene expressions across layers were examined by two statistical models: 1) The ‘Enrichment’ model. Layer-level summarized gene expression result was first fitted using the lmFit and eBayes function from the R package “limma” (version 3.16), after being blocked by the six pairs of spatially adjacent replicates and taking this correlation into account as computed by duplicateCorrelation. Then the Student’s t-test statistics was used to compare each layer against the other six using the layer-level data. This resulted in seven sets of Student’s t-test statistics (one per layer) with double-sided P values. We focused on genes with positive Student’s t-test statistics (expressed higher in one layer against the others) because these are enriched genes rather than depleted genes. 2) The ‘Pairwise’ model used the same “limma” functions for data processing and taking into account the same correlation structure in addition to using the contrasts.fit function provided by “limma”. Then we also computed the Student’s t-test statistics for each pair of layers. The Student’s t-test statistics with double-sided P values for both ‘Enrichment’ model and ‘Pairwise’ model were provided in Supplementary Table 3.

Below we showed the layer level differential expression statistics based on the ‘enrichment’ model. Our analysis showed that the SiGra enhanced data showed consistent results with the original study¹ (Maynard et al., 2021). Full details were provided in Supplementary Table 3.

Based on enhanced data							
	t_stat_WM	t_stat_Layer1	t_stat_Layer2	t_stat_Layer3	t_stat_Layer4	t_stat_Layer5	t_stat_Layer6
HPCAL1	-2.515415588	0.763348864	10.80973528	3.068050458	-2.740346896	-3.342613857	-1.006971103
KRT17	1.816446153	-2.616930966	-2.250056669	-3.502730791	-2.286232486	1.243251049	9.282056344
TRABD2A	-2.755949326	-1.582496736	-1.974004787	-2.089077109	1.065539817	13.09728313	-0.750503267
LAMP5	-3.676577066	1.353341098	7.26752965	2.67446215	-0.581769987	-2.015316293	-2.46167612
AQP4	3.9241445	5.809015348	-0.502858256	-2.228436465	-2.713284886	-1.93252492	-0.533701576
FREM3	-2.554736235	-0.429435062	4.048191737	6.161544982	-0.796887497	-1.810476872	-2.842155662
Based on raw data (from Table S4 in Maynard et al., 2021)							
	t_stat_WM	t_stat_Layer1	t_stat_Layer2	t_stat_Layer3	t_stat_Layer4	t_stat_Layer5	t_stat_Layer6
HPCAL1	-0.53214124	2.405065728	7.493447127	2.683311804	-4.081521288	-4.75055217	-0.845880986
KRT17	3.772136374	-2.453251903	-3.680479024	-4.132595657	-1.748237198	0.818820557	8.163120915
TRABD2A	-2.64674483	-1.02426376	-2.48564806	-1.32784384	1.636915534	8.198143596	-0.909222759
LAMP5	-3.995902629	2.932309431	6.740330171	2.926794549	0.253117241	-3.213691777	-3.376587143
AQP4	4.656646017	7.400647951	-1.009003858	-2.35501217	-3.696268588	-2.265381578	-0.115625687
FREM3	-5.132015759	1.190078713	3.637180776	5.369101063	0.20354803	-1.477185956	-2.815199561

2) SiGra identifies RELN40 in L1– but the cited paper seems to refer to temporal lobe and not DLPFC. Experimental validation is mandatory to validate these claims.

Response: Thanks for the reviewer’s comment. In the Maynard et al., 2021 (Extended Data Fig. 7), smFISH was used for validation of L1-marker gene RELN. We copied the figure here for the convenience of the reviewer. In the panel a, RELN was highly expressed in L1 in the 10x Visium data. In the second column in panel c, expression of RELN in L1 was validated by smFISH.

We also carefully compared the RELN gene in our enhanced data with the original study¹ (Maynard et al., 2021), which showed that RELN was strongly enriched in the Layer 1 in the enhanced data (Supplementary Table 3). Below was the layer level differential expression statistics of RELN based on the ‘enrichment’ model.

Based on enhanced data							
	t_stat_WM	t_stat_Layer1	t_stat_Layer2	t_stat_Layer3	t_stat_Layer4	t_stat_Layer5	t_stat_Layer6
RELN	-2.110680649	10.89659487	3.500784037	-0.501608148	-1.640299621	-1.759342543	-2.554834864
Based on raw data (from Table S4 in Maynard et al., 2021)							
	t_stat_WM	t_stat_Layer1	t_stat_Layer2	t_stat_Layer3	t_stat_Layer4	t_stat_Layer5	t_stat_Layer6
RELN	-2.636294906	9.62252324	3.096709323	0.381993222	-1.107599463	-1.774237554	-3.021050124

3) The authors show that e.g., BayesSpace misidentifies the neutrophile as lymphocytes, meanwhile, it incorrectly identifies some tumor cells as myeloid cells or neutrophile. Seurat fails to disentangle epithelial cells from tumor cells... What type of errors is SiGra making when compared to ground truth?

Response: Thanks for the reviewer's comment. When comparing with the provider-labeled cell types, the major difference is that some cells labeled as myeloid cells (Mcell) are in SiGra's neutrophil cluster. This is possibly due to that neutrophils belong to the myeloid cell lineage and thus the gene expression patterns of some myeloid cells are similar to neutrophils.

4) Is the extend of lipofuscin and autofluorescence affecting the performance of SiGra?

Response: Thanks for the reviewer's comment. Lipofuscin accumulates in brain tissues during aging or under pathologic conditions, and forms plaques of around $10\mu\text{m}^2$. Such lipofuscin plaques emit autofluorescence signals across major florescent channels used in single cell spatial images. To examine if the extend of lipofuscin and autofluorescence would affect the performance of SiGra, we randomly overlaid simulated autofluorescence signals from plaques of $10\mu\text{m}$ -by- $10\mu\text{m}$ to all channels in the original image data. The autofluorescence signal intensity was simulated by signals following normal distribution with mean as 10, 20, and 40, respectively. The figure below showed the zoomed-in figures of the images with added lipofuscin autofluorescence signals. The simulated lipofuscin autofluorescence slightly undermined SiGra's performance, from the original ARI (ARI: 0.55) to 0.513 and 0.509, with the added mild (mean signal: 10) or significant (mean signal: 20) autofluorescence signals, respectively. Of note, when the autofluorescence signals was overwhelming (mean signal of 40), the performance of SiGra dropped to 0.41. This simulation experiment suggested that under common experimental conditions, the lipofuscin autofluorescence or other types of autofluorescence would not significantly affect the SiGra performance.

Related revisions were made in our revised manuscript (page 7) and Supplementary Note 4.

5) Incidentally, I wonder whether SiGra could also help addressing inconsistencies in dissections (specifically considering gyrification of the human cortex). Could this even be a concern when using SiGra?

Response: Thanks for the reviewer's comment. The performance of SiGra is examined at different regions of the brain cortex gyrification. The 12 DLPFC slices spanned different gyrus regions. Specifically, the slices 151673, 151674, 151675, and 151676 represent both the gyrus and the sulcus regions, slides 151507, 151508, 151509, and 151510 represent the bottom of the sulcus region. SiGra achieves better performance than other methods in identifying the spatial domains across different dissections of the brain cortex gyrification.

6) In figure 6 only 2 of the 12 slices are shown. Please provide criteria for selecting these slices out of the 12.

Response: We appreciate the reviewer's comment. We provided all slices with their results below and also in our **Supplementary Fig. 3**.

Some additional minor points:

- All abbreviations should be explained when used first in the manuscript.

Response: Thanks for the reviewer's comment. We added full names and explained every abbreviation when first used in the manuscript.

- Reference 2 and 42 are identical

Response: Thanks for the reviewer's comment. We have revised the references accordingly.

References

1. Maynard, K.R., *et al.* Transcriptome-scale spatial gene expression in the human dorsolateral prefrontal cortex. *Nat Neurosci* **24**, 425-436 (2021).
2. Goodchild, M., Haining, R. & Wise, S. Integrating GIS and spatial data analysis: problems and possibilities. *International Journal of Geographical Information Systems* **6**, 407-423 (1992).
3. Shirowzhan, S. & Sepasgozar, S.M.E. Spatial Analysis Using Temporal Point Clouds in Advanced GIS: Methods for Ground Elevation Extraction in Slant Areas and Building Classifications. in *ISPRS International Journal of Geo-Information*, Vol. 8 (2019).
4. Park, J., *et al.* Cell segmentation-free inference of cell types from in situ transcriptomics data. *Nature Communications* **12**, 3545 (2021).
5. He, S., *et al.* High-plex imaging of RNA and proteins at subcellular resolution in fixed tissue by spatial molecular imaging. *Nat Biotechnol* **40**, 1794-1806 (2022).
6. Liu, J., *et al.* Concordance of MERFISH spatial transcriptomics with bulk and single-cell RNA sequencing. *Life Science Alliance* **6**, e202201701 (2023).
7. Single-cell transcriptomics of 20 mouse organs creates a Tabula Muris. *Nature* **562**, 367-372 (2018).
8. Almanzar, N., *et al.* A single-cell transcriptomic atlas characterizes ageing tissues in the mouse. *Nature* **583**, 590-595 (2020).
9. Vaswani, A., *et al.* Attention is All you Need. Vol. 30 (eds. Guyon, I., *et al.*) (2017).
10. Zhang, J., Zhang, H., Xia, C. & Sun, L. Graph-bert: Only attention is needed for learning graph representations. *arXiv preprint arXiv:2001.05140* (2020).
11. Yun, S., Jeong, M., Kim, R., Kang, J. & Kim, H.J. Graph transformer networks. *Advances in neural information processing systems* **32**(2019).
12. Zhao, H., Jiang, L., Jia, J., Torr, P.H.S. & Koltun, V. Point transformer. 16259-16268.
13. Dwivedi, V.P. & Bresson, X. A generalization of transformer networks to graphs. *arXiv preprint arXiv:2012.09699* (2020).
14. Shi, Y., *et al.* Masked label prediction: Unified message passing model for semi-supervised classification. *arXiv preprint arXiv:2009.03509* (2020).
15. Dwivedi, V.P., Luu, A.T., Laurent, T., Bengio, Y. & Bresson, X. Graph neural networks with learnable structural and positional representations. *arXiv preprint arXiv:2110.07875* (2021).
16. Bao, F., *et al.* Integrative spatial analysis of cell morphologies and transcriptional states with MUSE. *Nature Biotechnology* **40**, 1200-1209 (2022).
17. Furlan, M. Von Willebrand factor: molecular size and functional activity. *Annals of Hematology* **72**, 341-348 (1996).
18. Groeneveld, D.J., Poole, L.G. & Luyendyk, J.P. Targeting von Willebrand factor in liver diseases: A novel therapeutic strategy? *J Thromb Haemost* **19**, 1390-1408 (2021).
19. Ramadori, P., Klag, T., Malek, N.P. & Heikenwalder, M. Platelets in chronic liver disease, from bench to bedside. *JHEP Rep* **1**, 448-459 (2019).
20. Su, T., *et al.* Single-Cell Transcriptomics Reveals Zone-Specific Alterations of Liver Sinusoidal Endothelial Cells in Cirrhosis. *Cell Mol Gastroenterol Hepatol* **11**, 1139-1161 (2021).
21. Gray, D.A. & Woulfe, J. Lipofuscin and aging: a matter of toxic waste. *Sci Aging Knowledge Environ* **2005**, re1 (2005).

REVIEWER COMMENTS

Reviewer #1 (Remarks to the Author)

This revised version is improved with additional evaluations as compared to the original one. However, some main concerns still need to be addressed with satisfaction.

1. Regarding my previous comment 1, the authors provided an additional ablation study to test how different model structures affect the performance of SiGra. The ablation study is superficial as it only focuses on finding one setting that achieves the highest ARI for the benchmarking dataset. No insight into the model structure is provided. Since three autoencoders are used, how does the attention mechanism balance outputs from the three autoencoders? Is there any detected spatial domain defined by gene expression or image, or both? Is there any spatial domain that can only be revealed by combining gene expression and image? These questions are essential for users to understand the pros and cons of SiGra, but remain unanswered.

The ablation study focuses on selecting the model structure that has the highest ARI on the benchmarking data, which is problematic. Over-finetuning on a single dataset may lead to inaccurate conclusions. For example, when comparing the image-to-image autoencoder and image-to-gene autoencoder, the authors analyzed human dorsolateral prefrontal cortex data, and they concluded that the image-to-gene autoencoder is more informative than the image-to-gene autoencoder in spatial domain detection. I was not surprised to see that the image features play a less important role than gene expression in this dataset as the histology image in this dataset is not informative as other datasets - the brain layers cannot be clearly distinguished on the image. However, this conclusion does not necessarily hold for other datasets complemented by informative histology images, and the author may find image features more informative than gene expression.

2. Regarding my previous comment 2, the authors showed that the value of many parameters in SiGra (λ_1 , λ_2 , D_1 , D_2) are determined by a grid search approach. Different parameters are selected for data generated using different techniques to ensure the best performance in benchmarking. This approach limits the usefulness of SiGra as its parameter search requires the ground truth label for supervision. In addition, one set of parameters that achieves the highest ARI in the benchmarking dataset does not guarantee good performance on the others. The robustness of SiGra is a concern.

My overall feeling is that SiGra is a complex but ad hoc model with a number of changeable parameters. The model's performance is sensitive to the choice of these parameters, and the values of these parameters are determined by a cherry-picking strategy using the benchmarking dataset with limited scientific consideration. This limitation makes SiGra hard to use in practice.

3. Regarding my previous comment 4, the response is not satisfactory and dismissive. "Cell type" and "spatial region" are two distinct concepts. It is incorrect and misleading to call "cell types" "spatial regions at the cellular level." I insist that comparing the predicted spatial domains from SiGra and other methods using annotated cell types as the ground truth is incorrect.

The authors also used a moving window to transform SiGra's cell type prediction into spatial regions and compared it with other methods for spatial region detection. This is an unfair comparison because 1) other methods are developed for spot-level ST data while single-cell data are provided as input; 2) the moving window agglomeration approach is only used in SiGra while other methods are not. I do not think this NanoString dataset is suitable for evaluating spatial region detection.

4. Regarding my previous comment 6, the evaluation of L-R is still not convincing. No concrete evidence is provided to show whether the L-R association detected from SiGra-enhanced data are true biological interactions or not. The authors only showed more L-R can be identified using enhanced data, and the enhanced-specific LR has slightly smaller FDRs than the raw-specific L-R,

although they are all significant. However, smaller FED does not necessarily mean the SiGra-enhanced data can better identify true biological interactions. More is not always better, and this is the core of my criticism: a "biased" method can amplify a signal in a biased way under the hood, leading to any number of downstream discoveries.

Reviewer #2 (Remarks to the Author)

The authors have addressed most of my concerns, but there are still some minor concerns as follows:

1. Regarding the terminology of `\textit{graph transformer convolutional layer}`, I respect the authors interpretation. However, based on our understanding, the main difference between Sigra's "graph transformer" and graph attention networks lies in the attention mechanism. Sigra uses multiplicative attention, while GAT uses concat attention, which is a variant of additive attention. However, as mentioned in a survey \cite{niu2021review}, concat attention has been shown to be more effective than multiplicative attention in several studies, despite its higher computational cost. However, GAT chooses concat attention because the sparse structure of graphs allows for a performance-computation trade-off. Therefore, it is reasonable to conclude that GAT is a highly specialized model for graph data and is extremely similar to Sigra. Due to the similarity and complete of literature, I will recommend to cite GAT.

2. The claim of utilizing 'multi-channel images of cell morphology...' is kind of overstated, because vectorizing the image with fixed size containing multiple cells and possible some holes will not guarantee to learn and capture the morphology information. More evidences, such as the shape or other related morphology features, need to be shown to support this claim.

Reviewer #3 (Remarks to the Author)

The authors have made great effort in answering my questions and concerns from the previous round of review. My questions have been well resolved.

RESPONSE TO REVIEWERS' COMMENTS

Reviewer #1

This revised version is improved with additional evaluations as compared to the original one. However, some main concerns still need to be addressed with satisfaction.

1. Regarding my previous comment 1, the authors provided an additional ablation study to test how different model structures affect the performance of SiGra. The ablation study is superficial as it only focuses on finding one setting that achieves the highest ARI for the benchmarking dataset. No insight into the model structure is provided. Since three autoencoders are used, how does the attention mechanism balance outputs from the three autoencoders? Is there any detected spatial domain defined by gene expression or image, or both?

Is there any spatial domain that can only be revealed by combining gene expression and image? These questions are essential for users to understand the pros and cons of SiGra, but remain unanswered.

Response: Thanks for the reviewer's suggestions. In our revision, we performed structural ablation analysis and visualized the results (Supplementary Fig. 7b). These results provide compelling evidence that our model's architecture is well designed, with each component serving a crucial role and exhibiting a synergistic effect. Notably, removing any component from the model leads to a significant decline in performance, further emphasizing the indispensability of each element.

Besides the ARI score, we further illustrate the contributions of the three encoder-decoders to the detected spatial domain using slice 151507 (Fig. 5b). As demonstrated in the following figure:

- The image-to-image auto-encoder (I-AE) alone can only partially identify the Layer 1 and the White Matter region. The Layer 2/3,4,5,6 region is false detected as two major regions (left: purple; right: green).
- The image-to-gene encoder-decoder (I-ED) provides more accurate detection in Layer 1 and White Matter region. However, the two falsely detected regions by the I-AE model also appear here.
- The gene-to-gene encoder-decoder (G-ED) model detects most layers, with the Layer 4 and the Layer 6 missing.

- The hybrid encoder-decoder (H-ED) model partially identifies the Layer 2, but still missed the Layer 4 and the Layer 6. However, H-ED falsely detects three regions (labeled as two thin red regions adjacent to Layer 1). Meanwhile, the Layer 2 region is detected without clear boundaries.
- The SiGra model successfully identifies all regions except a small Layer3 region on the top-right corner. Moreover, the regions Layer 4 and Layer 6 can only be detected in SiGra.

These results suggest that the superior performance of the SiGra model is not due to the linear addition of the contributions of the image-based models (I-ED) and the gene-based models (G-ED). The simple integration of the image and the gene information by the H-ED model does not achieve better performance. Meanwhile, in the SiGra model, the image information and the gene expression information demonstrate strong synergy: spatial domains (Layer 4 and Layer 6) that are not identified in either the image-based model (I-AE and I-ED), or the gene-based models (G-ED), or the hybrid model (I-ED) can be identified by the SiGra model.

These results further confirm the structure and architecture of the SiGra model is carefully designed. Missing any component (such as I-ED, G-ED, or H-ED) will dramatically undermine the performance of the SiGra model.

The ablation study focuses on selecting the model structure that has the highest ARI on the benchmarking data, which is problematic. Over-finetuning on a single dataset may lead to inaccurate conclusions. For example, when comparing the image-to-image autoencoder and image-to-gene autoencoder, the authors analyzed human dorsolateral prefrontal cortex data, and they concluded that the image-to-gene autoencoder is more informative than the image to-gene autoencoder in spatial domain detection. I was not surprised to see that the image features play a less important role than gene expression in this dataset as the histology image in this dataset is not informative as other datasets - the brain layers cannot be a clearly distinguished on the image. However, this conclusion does not necessarily hold for other datasets complemented by informative histology images, and the author may find image features more informative than gene expression.

Response: Thanks for the reviewer's comments. Here we provide the ablation results on additional new datasets.

To investigate whether the model structure is over finetuned on DLPFC tissues, we further perform ablation analysis on additional datasets. Three 10x Visium datasets from cancer tissues are used, with pathologist's annotation as ground truth. These datasets are: one prostate cancer (<https://www.10xgenomics.com/resources/datasets/human-prostate-cancer-adenocarcinoma-with-invasive-carcinoma-ffpe-1-standard-1-3-0>) and two breast cancer tissue slices (BRCA-1, <https://www.10xgenomics.com/resources/datasets/human-breast-cancer-ductal-carcinoma-in-situ-invasive-carcinoma-ffpe-1-standard-1-3-0>; BRCA-2, <https://www.10xgenomics.com/resources/datasets/human-breast-cancer-block-a-section-1-1-standard-1-0-0>). In these tumor slices, the histology images may be more informative for spatial domains than DLPFC. Our results below further suggest that the SiGra model outperforms other designs on cancer datasets.

- Prostate cancer 10x Visium dataset

- BRCA-1 10x Visium dataset

- BRCA-2 10x Visium dataset

Our results indicate that the SiGra model remains the optimal and robust design on these cancer datasets. In contrast, the performance of ablated models is dataset specific. Specifically, for prostate cancer and BRCA-2, as the reviewer expected, images are more informative. Comparing with the gene-only models (G-ED, with ARIs of 0.38 and 0.16, respectively), the models that involve both image and gene show better performance (H-ED: 0.49 and 0.32; I-ED: 0.44 and 0.31). For the BRAC-2 dataset, the image autoencoder (I-AE: ARI 0.28) outperforms the gene autoencoder (G-ED: ARI: 0.16). Meanwhile, on the BRAC-1 dataset, genes (G-ED: 0.35) contribute more than images (I-AE: 0.05, I-ED: 0.32) or the hybrid models (H-ED: 0.23).

In summary, the SiGra model is a robust design that outperforms ablated models. The contribution of images and genes as well as the performance of other ablated models is dataset specific. These ablation results further support the generalizability of the SiGra model in identifying spatial domains.

2. Regarding my previous comment 2, the authors showed that the value of many parameters in SiGra (λ_1 , λ_2 , D1, D2) are determined by a grid search approach. Different parameters are selected for data generated using different techniques to ensure the best performance in benchmarking. This approach limits the usefulness of SiGra as its parameter search requires the ground truth label for supervision. In addition, one set of parameters that achieves the highest ARI in the benchmarking dataset does not guarantee good performance on the others. The robustness of SiGra is a concern.

My overall feeling is that SiGra is a complex but ad hoc model with a number of changeable parameters. The model's performance is sensitive to the choice of these parameters, and the values of these parameters are determined by a cherry-picking strategy using the benchmarking dataset with limited scientific consideration. This limitation makes SiGra hard to use in practice.

Response: Thanks. We would like to address the reviewer's comments in four aspects below.

1) Generalizability and robustness of SiGra on additional datasets.

We further demonstrated the robustness of the SiGra model on additional datasets with default hyperparameters. That is, users do not need to finetune the hyperparameters for their own datasets.

Three 10x Visium datasets from prostate cancer (<https://www.10xgenomics.com/resources/datasets/human-prostate-cancer-adenocarcinoma-with-invasive-carcinoma-ffpe-1-standard-1-3-0>) and breast cancer tissue slices (BRCA1, <https://www.10xgenomics.com/resources/datasets/human-breast-cancer-ductal-carcinoma-in-situ-invasive-carcinoma-ffpe-1-standard-1-3-0>; BRCA2, <https://www.10xgenomics.com/resources/datasets/human-breast-cancer-block-a-section-1-1-standard-1-0-0>) obtained from 10x Genomics website, are used to benchmark the performance of SiGra and competitors, with pathologist's annotation as ground truth.

- Prostate cancer 10x Visium dataset

Three regions (tumor tissue, prostate tissue, and prostatic stroma) are identified and annotated by pathologist. SiGra (ARI: 0.70) outperforms the other four methods (Scanpy: 0.36, SpaGCN: 0.40, stLearn: 0.52, Seurat: 0.48, and BayesSpace: 0.38).

- BRCA-1 10x Visium dataset

Four regions (tumor tissue, desmoplastic region, lymphocyte-enriched region, and the necrosis and hemorrhage region) are identified and annotated by pathologist. SigrA (ARI: 0.59) outperforms the other four methods (Scanpy: 0.55, SpaGCN: 0.48, stLearn: 0.38, Seurat: 0.57, and BayesSpace: 0.37).

- BRCA-2 10x Visium dataset

Three regions (tumor region, desmoplastic region, and lymphocyte-enriched region) are identified and annotated by pathologist. Sigra (ARI: 0.42) outperforms the other four methods (Scanpy: -0.10, SpaGCN: 0.15, stLearn: 0.29, Seurat: -0.09, and BayesSpace: -0.09). Codes for reproducing the results are available as Jupyter notebook at <https://github.com/QSong-github/SiGra/tree/main/Tutorials>.

Our results demonstrate that the SiGra model can be directly used on new datasets with default hyperparameters, and the performance is superior to competitors. Meanwhile, the SiGra model supports dataset-specific finetuning of the hyperparameters. This function is optional.

2) Hyperparameter tuning does not use ground truth labels and does not use supervision.

We would like to re-emphasize that the ground truth labels are **not** used in the hyperparameter search. Instead, the **loss** on the validation set (30% of the overall data) is used for choosing the hyperparameters (Supplementary Note 5). Here we re-post the text from the previous Response Letter with the corresponding contents highlighted:

...

We first performed coarse searches to identify the optimal parameter range for each data type, then used a grid-search approach for fine-tuning to determine the optimal values. The best options for the two parameters were chosen based on the loss evaluation on the validation set (30% of the overall data).

...

Similar to the selection of λ_1 and λ_2 , we fine-tuned the dimensions (D_1 and D_2) of the 1st and 2nd layers respectively, based on the loss obtained from the validation set (30% of the overall data).

...

3) The number of hyperparameters of SiGra model is modest.

The complexity of SiGra is modest. Below we have compared the number of hyperparameters of SiGra with other state-of-art deep learning models for analyzing spatial and single-cell data. The comparisons are summarized in the following table:

Model	Number of hyperparameters	Hyperparameters	Journal and Years
SiGra	4	$\lambda_1, \lambda_2, D_1, D_2$	N/A
STAGATE ¹	4	Encoder layer number, Decoder layer number, latent dimension, the weight of cell type aware SNN	Nature Communications, 2022
SpaGCN ²	4	Scaling parameter s , characteristic length scale l , area of each spot b , percentage of contribution from neighborhoods p	Nature Methods, 2021
scGNN ³	10	Weights in loss: $\alpha, \beta, \gamma_1, \gamma_2$; k-neighbors, LTMG intensity, type of LTMG, graph embedding type,	Nature Communications, 2021

		clustering methods, use of autoencoder	
MUSE ⁴	5	$\lambda_{regularization}$, $\lambda_{supervise}$, latent dimension, cluster update interval, n neighbors	Nature Biotechnology, 2022

4) Justifications of the model structure.

The final architecture of the SiGra model was built through thoughtful designs and thorough testing of other optional architectures. In our original manuscript, we did not include the testing results that justify the current architecture, as well as the results for hyperparameters tuning. We would like to express our gratitude to the reviewer for highlighting the importance of presenting technical results. It is indeed crucial for the audience to comprehend the process of determining the final architecture and selecting the appropriate hyperparameters. After our 1st and 2nd revision, we hope that we have provided sufficient technical results to demonstrate that 1) the model architecture is not ad hoc designed but is built through thorough and holistic testing; 2) each component significantly improves the model performance and thus necessary; 3) SiGra outperforms competitors on the other additional datasets with hyperparameters determined by the previous benchmarking datasets, therefore, users do not necessarily have to further tune hyperparameters when applying SiGra to their new datasets; 4) both the SiGra architecture and hyperparameters are examined on new datasets, which demonstrate robust outperformance.

3. Regarding my previous comment 4, the response is not satisfactory and dismissive. "Cell type" and "spatial region" are two distinct concepts. It is incorrect and misleading to call "cell types" "spatial regions at the cellular level." I insist that comparing the predicted spatial domains from SiGra and other methods using annotated cell types as the ground truth is incorrect.

Response: Thanks. We would like to address the reviewer's comments in three aspects below.

1) Benchmarking methods for cell identity identification

Most methods we chose for comparison have officially claimed that they can be used for detecting cell identities from single-cell ST data. Therefore, comparing SiGra with other methods using annotated cell types as the ground truth is fair. Specifically,

Seurat pipeline was not initially developed for spot-level ST data, but for single-cell RNA-seq data. They have officially claimed that their pipeline can be used for both spot-level ST data (https://satijalab.org/seurat/articles/spatial_vignette.html) and single-cell ST data (https://satijalab.org/seurat/articles/spatial_vignette_2.html). The following figures are from Seurat's vignette.

Spatial distribution of cell Identities

Cell Identities

Markers

[SpaGCN](https://github.com/jianhuupenn/SpaGCN) claimed that it can be used for single-cell ST data too (MERFISH data, <https://github.com/jianhuupenn/SpaGCN>).

“SpaGCN is applicable to both **in-situ transcriptomics with single-cell resolution** (seqFISH, seqFISH+, **MERFISH**, STARmap, and FISSEQ) and spatial barcoding-based transcriptomics (Spatial Transcriptomics, SLIDE-seq, SLIDE-seqV2, HDST, 10x Visium, DBiT-seq, Stereo-seq, and PIXEL-seq) data.”

[stLearn](https://stlearn.readthedocs.io/en/latest/tutorials/Xenium_PSTS.html) also made such claim and was applied to the 10x Xenium data (https://stlearn.readthedocs.io/en/latest/tutorials/Xenium_PSTS.html). The follow figures are from its tutorial:

Spatial distributions of cell identities

[Scanpy](https://scanpy-tutorials.readthedocs.io/en/latest/spatial/basic-analysis.html#MERFISH-example) also released their official example for single-cell ST data (MERFISH data, <https://scanpy-tutorials.readthedocs.io/en/latest/spatial/basic-analysis.html#MERFISH-example>).

Cell Identities

Markers

2) Clarification on spatial domain detection

We respect the reviewer’s opinion and have modified our descriptions accordingly to avoid potential controversies or confusions. Specifically, we added the following in **Materials and Methods** (page 9):

“Spatial domain detection

1) For the spot-level spatial data that has a low spatial resolution and consists of mixed cells/cell types in each spot, SiGra directly detects the spatial domains by clustering the latent-represented spots using Leiden. 2) For single-cell spatial data, SiGra first identifies the cell types for each individual cell by clustering the latent-representation using Leiden, and then reveals spatial domains via a dimensional moving window agglomeration approach⁵. Specifically, the spatially distributed cells are summarized by a circular window of diameter d sliding at both x and y directions across the whole image with a given stride length s . At each stop $C_{i,j}$ with the coordinate (x_i, y_j) , a vector $c_{i,j} \equiv [q_1, \dots, q_t]$ representing the proportions of the SiGra identified clusters (t) covered by the sliding window is calculated. All the stops $\{C_{i,j}\}$ are recursively merged to k groups $\{a_1, \dots, a_k\}$ by hierarchical clustering according to the cluster proportion vectors $\{c_{i,j}\}$. These agglomerated groups are defined as spatial domains. The window radius d used in **Supplementary Fig. 4** is $100\mu\text{m}$, which is consistent with the $10\times$ Visium spatial resolution, with the stride s of $10\mu\text{m}$. For fair comparisons, the same moving window agglomeration approach is used in benchmarking methods (Supplementary Fig. 4). The ground truth of the anatomic spatial domains for the DLPFC slices and lung cancer slices is obtained from the original study⁶ and the certificated pathologist at Indiana University Health (T.H.).”

3) Comparisons with benchmarking methods using spatial domain

Moreover, since the reviewer does not agree on utilizing cell types as the evaluation criterion, we also compared the performance based on pathologist's annotation. Based on our previous 1st revision, as shown in Supplementary Fig. 4, three spatial domains were identified by pathologist: the tumor region (green), the desmoplasia region (red), and the adjacent normal region (orange). For fair comparisons with other methods, the same moving window agglomeration approach was used. Compared with the ground truth, SiGra achieved an ARI of 0.60, better than other methods including BayesSpace (ARI: 0.25), SpaGCN (ARI: 0.10), Seurat (ARI: 0.10), stLearn (ARI: 0.10), and Scanpy (ARI: 0.17). These results showed that SiGra obtained reliable spatial domains based on its identified accurate cell identities. It also indicated that the NanoString CosMx profiled cancer tissue slice was much more challenging given its strong cellular heterogeneity, large cell number, and high-resolution, compared with the 10x Visium profiled normal DLPFC tissues which have well-organized anatomic structure.

To further verify the comparison results, in our previous 1st revision, we also tested BayesSpace and SpaGCN for direct spatial domain identification of the three domains, without using the moving window agglomeration approach. BayesSpace and SpaGCN only obtained ARIs of 0.15 and 0.19, respectively. These results supported that, when detecting large-scale anatomic spatial domains from single-cell spatial data, it was necessary to agglomerate the high-resolution cellular-level clustering results.

The authors also used a moving window to transform SiGra's cell type prediction into spatial regions and compared it with other methods for spatial region detection. This is an unfair comparison because 1) other methods are developed for spot-level ST data while single-cell data are provided as input; 2) the moving window agglomeration approach is only used in SiGra while other methods are not. I do not think this NanoString dataset is suitable for evaluating spatial region detection.

Response: Thanks. We would like to address the reviewer's comments in three aspects as below.

1) Fairness in comparing with methods that were originally developed for spot-level ST data.

Most methods we chose for comparison have recently and officially claimed that they can be used for single cell ST data. For example, **Seurat** pipeline has officially claimed that their pipeline can be used for both spot-level ST data (https://satijalab.org/seurat/articles/spatial_vignette.html) and the single-cell ST data (https://satijalab.org/seurat/articles/spatial_vignette_2.html). **SpaGCN** claimed that it can be used for single-cell ST data too (MERFISH data, <https://github.com/jianhuupenn/SpaGCN>), and claims that:

“SpaGCN is applicable to both **in-situ transcriptomics with single-cell resolution** (seqFISH, seqFISH+, **MERFISH**, STARmap, and FISSEQ) and spatial barcoding-based transcriptomics (Spatial Transcriptomics, SLIDE-seq, SLIDE-seqV2, HDST, 10x Visium, DBiT-seq, Stereo-seq, and PIXEL-seq) data.”

stLearn also made such claim (https://stlearn.readthedocs.io/en/latest/tutorials/Xenium_PSTS.html). **Scanpy** released the official example for single-cell ST data (MERFISH data, <https://scanpy-tutorials.readthedocs.io/en/latest/spatial/basic-analysis.html#MERFISH-example>). Only BayesSpace has not yet release such claim. Therefore, the comparisons between SiGra and these chosen methods are fair.

2) Moving window was used for other methods too.

In our previous 1st revision, we stated that the moving window agglomeration was used for other methods too. When using moving window for the other methods, Supplementary Fig. 4 shows that SiGra outperformed competitors.

Meanwhile, we also tested BayesSpace and SpaGCN for direct spatial domain identification of the three domains, without using the moving window agglomeration approach. BayesSpace and SpaGCN only obtained ARIs of 0.15 and 0.19, respectively, comparing with the ARI of 0.25 and 0.10 (with moving window approach). These results further demonstrated that, for detecting large-scale anatomic spatial domains from single-cell spatial data, it was necessary to agglomerate the high-resolution cellular-level clustering results.

3) SiGra achieves consistent results with pathological annotations on single-cell ST data.

We respect the opinion of the reviewer that single-cell ST data such as the NanoString CosMx SMI data is not suitable for evaluating spatial region detection. However, we have different opinions. First, single-cell ST data provides gene expression information at a higher spatial resolution than spot-level ST data. Therefore, if spatial regions can be detected in low-resolution data, they should also be detected in high-resolution data. Second, there are real-world needs in identifying spatial regions on single-cell ST data too, and SiGra provides such functionality. Therefore, evaluating the performance of SiGra on such data can provide meaningful information to end users. Third, our results on NanoString CosMx SMI data show that the spatial regions identified by SiGra are accurate based on the pathologist's annotation (ARI: 0.60), which demonstrates its capability to identify spatial regions on single-cell ST data. In summary, single-cell ST data with sufficient information can also be used for spatial region detection.

4. Regarding my previous comment 6, the evaluation of L-R is still not convincing. No concrete evidence is provided to show whether the L-R association detected from SiGra-enhanced data are true biological interactions or not. The authors only showed more L-R can be identified using enhanced data, and the enhanced-specific LR has slightly smaller FDRs than the raw-specific L-R, although they are all significant. However, smaller FED does not necessarily mean the SiGra-enhanced data can better identify true biological interactions. More is not always better, and this is the core of my criticism: a "biased" method can amplify a signal in a biased way under the hood, leading to any number of downstream discoveries.

Response: Thanks for the reviewer's comments.

We agree with the reviewer that more is not always better, due to the potential false discoveries (false positive L-R associations). Therefore, we perform false discovery analysis to address this concern. The goal is to examine whether L-R associations can be more reliably identified from the enhanced data. The goal is not to experimentally check all possible L-R associations, generate the ground truth of positive and negative L-R associations, and then estimate how many true L-R associations are missed (type-II error) and how many false L-R associations are mistakenly detected as true associations (type-I error). Practically, this is difficult for most labs.

When there is no such ideal ground truth available, model performance can still be reliably compared by permutation tests, since the permuted data can be generated to estimate the distribution of the false L-R associations. This statistical approach has been widely used in bioinformatics^{7,8}. The false discovery analysis suggests that SiGra-enhanced data allows detecting L-R associations at a lower false discovery rate.

Moreover, we also show that the L-R associations detected from SiGra-enhanced data are also observed in additional dataset of the same tissue type.

Specifically, for the Vizgen MERSCOPE mouse liver dataset, as shown in Fig. 4, the y-axis and x-axis refer to the FDR values of each L-R pair in the enhanced and raw data respectively. Among the 64 L-R pairs identified in this dataset, 13 L-R pairs from the enhanced data are statistically significant ($FDR < 0.05$), whereas 12 L-R pairs from the raw data have $FDR < 0.05$. There are 9 L-R pairs shared between the enhanced and the raw data, indicating the enhanced data preserves useful information of the raw data. In addition, enhanced data has 4 specific L-R interactions, while raw data has 3 specific L-R interactions. We further examine the L-R interactions specifically identified from the enhanced and the raw data, respectively, using the bulk RNA-seq data from the Tabula Muris Consortium⁹ as the validation dataset. The 4 L-R pairs specifically identified from the SiGra's enhanced data also present strong correlations in the validation dataset ($Wnt2-Fzd4$: 0.581; $Pkm-Cd44$: 0.885; $Col1a2-Itga2b$: 0.641; $Dll1-Notch2$: 0.798). However, the raw-specific L-R pairs show low correlations in bulk data ($Fgf1-Egfr$: 0.386; $Timp3-Kdr$: 0.498; $Jag1-Notch1$: 0.115). These results suggest that the enhanced data specific L-R pairs are more likely to be true discoveries.

Moreover, to show the SiGra's enhanced data provide useful rather than bias information, based on the results of DLPFC (Fig. 5), we have carefully compared the layer-enriched gene markers ($HPCAL1$, $KRT17$, $TRABD2A$, $LAMP5$, $AQP4$, $FREM3$) in our enhanced data with the original study⁶ (Maynard et al., 2021). That is, we perform the exact statistical analysis in Maynard et al 2021 ("Layer-level gene modeling" and fit 'Enrichment' and 'Pairwise' models) using the enhanced data obtained by SiGra. The variations of gene expressions across layers are examined by two statistical models: **1)** The 'Enrichment' model. Layer-level summarized gene expression result is first fitted using the `lmFit` and `eBayes` function from the R package "limma" (version 3.16), after being blocked by the six pairs of spatially adjacent replicates and taking this correlation into account as computed by `duplicateCorrelation`. Then the Student's t-test statistics is used to compare each layer against the other six using the layer-level data. This result in seven sets of Student's t-test statistics (one per layer) with double-sided P values. We focus on genes with positive Student's t-test statistics (expressed higher in one layer against the others) because these are enriched genes rather than depleted genes. **2)** The 'Pairwise' model used the same "limma" functions for data processing and taking into account the same correlation structure in addition to using the `contrasts.fit` function provided by "limma". Then we also compute the Student's t-test statistics for each pair of layers. The Student's t-test statistics with double-sided P values for both 'Enrichment' model and 'Pairwise' model are provided in Supplementary Table 3.

Below is the layer level differential expression statistics based on the 'enrichment' model. Our analysis shows that the SiGra enhanced data highlights the layer-enriched markers for each brain layer, which is also consistent with the original study⁶ (Maynard et al., 2021).

Based on enhanced data							
	t_stat_WM	t_stat_Layer1	t_stat_Layer2	t_stat_Layer3	t_stat_Layer4	t_stat_Layer5	t_stat_Layer6
HPCAL1	-2.515415588	0.763348864	10.80973528	3.068050458	-2.740346896	-3.342613857	-1.006971103
KRT17	1.816446153	-2.616930966	-2.250056669	-3.502730791	-2.286232486	1.243251049	9.282056344
TRABD2A	-2.755949326	-1.582496736	-1.974004787	-2.089077109	1.065539817	13.09728313	-0.750503267
LAMP5	-3.676577066	1.353341098	7.26752965	2.67446215	-0.581769987	-2.015316293	-2.46167612
AQP4	3.9241445	5.809015348	-0.502858256	-2.228436465	-2.713284886	-1.93252492	-0.533701576
FREM3	-2.554736235	-0.429435062	4.048191737	6.161544982	-0.796887497	-1.810476872	-2.842155662
Based on raw data (from Table S4 in Maynard et al., 2021)							
	t_stat_WM	t_stat_Layer1	t_stat_Layer2	t_stat_Layer3	t_stat_Layer4	t_stat_Layer5	t_stat_Layer6
HPCAL1	-0.53214124	2.405065728	7.493447127	2.683311804	-4.081521288	-4.75055217	-0.845880986
KRT17	3.772136374	-2.453251903	-3.680479024	-4.132595657	-1.748237198	0.818820557	8.163120915
TRABD2A	-2.64674483	-1.02426376	-2.48564806	-1.32784384	1.636915534	8.198143596	-0.909222759
LAMP5	-3.995902629	2.932309431	6.740330171	2.926794549	0.253117241	-3.213691777	-3.376587143
AQP4	4.656646017	7.400647951	-1.009003858	-2.35501217	-3.696268588	-2.265381578	-0.115625687
FREM3	-5.132015759	1.190078713	3.637180776	5.369101063	0.20354803	-1.477185956	-2.815199561

Collectively, from the aspects of L-R pairs and DEG analysis, the SiGra-enhanced data provides more biological meaningful information than raw data.

Reviewer #2

The authors have addressed most of my concerns, but there are still some minor concerns as follows:

1. Regarding the terminology of \textit{graph transformer convolutional layer}, I respect the authors' interpretation. However, based on our understanding, the main difference between Sigra's "graph transformer" and graph attention networks lies in the attention mechanism. Sigra uses multiplicative attention, while GAT uses concat attention, which is a variant of additive attention. However, as mentioned in a survey \cite{niu2021review}, concat attention has been shown to be more effective than multiplicative attention in several studies, despite its higher computational cost. However, GAT chooses concat attention because the sparse structure of graphs allows for a performance-computation trade-off. Therefore, it is reasonable to conclude that GAT is a highly specialized model for graph data and is extremely similar to Sigra. Due to the similarity and completeness of literature, I will recommend to cite GAT.

Response: Thanks for the reviewer's comments. We agree with the reviewer's opinion and have cited the GAT paper in our revised version. We copied the revised text (page 2) here for the reviewer's convenience.

“In addition to domain recognition, the enhancement of spatial gene expression data also presents a significant challenge. Though great progress has been made in spatial technologies, the major problems such as missing values, data sparsity, low coverage, and noises^{2,15} encountered in spatial transcriptomics profiles are impeding the effective use and the elucidation of biology insights. Meanwhile, the multi-channel spatial images in single-cell spatial data consist of high-resolution, high-content features detected in the tissue, such as cell types, functions, and morphologies of cellular compartments, as well as the spatial distributions of cells. Incorporating such imaging features with cell-level transcriptomics data will help address the challenges of missing values and data noise. Moreover, as the spatial relations between an individual cell and its neighboring cells can be naturally represented as a spatial adjacency graph, graph-based artificial intelligence is promising for spatial data modeling. Notably, graph-based models enhanced with attention mechanisms¹⁰, such as the Graph Attention Network (GAT) and the graph convolutional transformer models^{11,12}, have demonstrated remarkable advancements and yielded significantly improved outcomes.”

2. The claim of utilizing 'multi-channel images of cell morphology...' is kind of overstated, because vectorizing the image with fixed size containing multiple cells and possibly some holes will not guarantee to learn and capture the morphology information. More evidences, such as the shape or other related morphology features, need to be shown to support this claim.

Response: Thanks for the reviewer's comments. We agree that our claim of utilizing 'multi-channel images of cell morphology' is overstated. In our revised manuscript (page 2 and page 7), we modified the specific text as “multi-channel images of cells and their niches”.

Reviewer #3

The authors have made great effort in answering my questions and concerns from the previous round of review. My questions have been well resolved.

Response: Thanks. We appreciate the reviewer's comments.

References

- 1 Dong, K. & Zhang, S. Deciphering spatial domains from spatially resolved transcriptomics with an adaptive graph attention auto-encoder. *Nature Communications* **13**, 1739, doi:10.1038/s41467-022-29439-6 (2022).
- 2 Hu, J. *et al.* SpaGCN: Integrating gene expression, spatial location and histology to identify spatial domains and spatially variable genes by graph convolutional network. *Nature Methods* **18**, 1342-1351, doi:10.1038/s41592-021-01255-8 (2021).
- 3 Wang, J. *et al.* scGNN is a novel graph neural network framework for single-cell RNA-Seq analyses. *Nature Communications* **12**, 1882, doi:10.1038/s41467-021-22197-x (2021).
- 4 Bao, F. *et al.* Integrative spatial analysis of cell morphologies and transcriptional states with MUSE. *Nature Biotechnology* **40**, 1200-1209, doi:10.1038/s41587-022-01251-z (2022).
- 5 Park, J. *et al.* Cell segmentation-free inference of cell types from in situ transcriptomics data. *Nature Communications* **12**, 3545, doi:10.1038/s41467-021-23807-4 (2021).
- 6 Maynard, K. R. *et al.* Transcriptome-scale spatial gene expression in the human dorsolateral prefrontal cortex. *Nat Neurosci* **24**, 425-436, doi:10.1038/s41593-020-00787-0 (2021).
- 7 Karasaki, T. *et al.* Evolutionary characterization of lung adenocarcinoma morphology in TRACERx. *Nature Medicine* **29**, 833-845, doi:10.1038/s41591-023-02230-w (2023).
- 8 Schweiger, R. *et al.* Detecting heritable phenotypes without a model using fast permutation testing for heritability and set-tests. *Nature Communications* **9**, 4919, doi:10.1038/s41467-018-07276-w (2018).
- 9 Single-cell transcriptomics of 20 mouse organs creates a Tabula Muris. *Nature* **562**, 367-372, doi:10.1038/s41586-018-0590-4 (2018).
- 10 Niu, Z., Zhong, G. & Yu, H. A review on the attention mechanism of deep learning. *Neurocomputing* **452**, 48-62, doi:<https://doi.org/10.1016/j.neucom.2021.03.091> (2021).
- 11 Yun, S., Jeong, M., Kim, R., Kang, J. & Kim, H. J. Graph transformer networks. *Advances in neural information processing systems* **32** (2019).
- 12 Shi, Y. *et al.* Masked label prediction: Unified message passing model for semi-supervised classification. *arXiv preprint arXiv:2009.03509* (2020).

REVIEWERS' COMMENTS

Reviewer #1 (Remarks to the Author):

My questions have been well addressed.

RESPONSE TO REVIEWERS' COMMENTS

Reviewer #1 (Remarks to the Author):

My questions have been well addressed.

Response: Thanks. We appreciate the reviewer's comment.